# A small molecule promotes cartilage extracellular matrix generation and inhibits osteoarthritis development

Yuanyuan Shi[1,2], Xiaoqing Hu[1,2], Jin Cheng[1], Xin Zhang[1], Fengyuan Zhao[1], Weili Shi[1], Bo Ren[1], Huilei Yu[1], Peng Yang[1], Zong Li[1], Qiang Liu[1], Zhenlong Liu [1], Xiaoning Duan[1], Xin Fu[1], Jiying Zhang[1], Jianquan Wang[1] & Yingfang Ao[1]

Degradation of extracellular matrix (ECM) underlies loss of cartilage tissue in osteoarthritis, a common disease for which no effective disease-modifying therapy currently exists. Here we describe BNTA, a small molecule with ECM modulatory properties. BNTA promotes generation of ECM components in cultured chondrocytes isolated from individuals with osteoarthritis. In human osteoarthritic cartilage explants, BNTA treatment stimulates expression of ECM components while suppressing inflammatory mediators. Intra-articular injection of BNTA delays the disease progression in a trauma-induced rat model of osteoarthritis. Furthermore, we identify superoxide dismutase 3 (SOD3) as a mediator of BNTA activity. BNTA induces SOD3 expression and superoxide anion elimination in osteoarthritic chondrocyte culture, and ectopic SOD3 expression recapitulates the effect of BNTA on ECM biosynthesis. These observations identify SOD3 as a relevant drug target, and BNTA as a potential therapeutic agent in osteoarthritis.

[1] Institute of Sports Medicine, Beijing Key Laboratory of Sports Injuries, Peking University Third Hospital, 100191 Beijing, China. [2] These authors contributed equally: Yuanyuan Shi, Xiaoqing Hu. Correspondence and requests for materials should be addressed to J.W. (email: wjqsportsmed@163.com) or to Y.A. (email: aoyingfang@163.com)

Osteoarthritis (OA) is the most prevalent musculoskeletal disorder, which is a major cause of disability and reduced quality of life in the elderly[1–4]. However, present therapeutic options for OA are predominantly palliative, such as pain management with analgesics and anti-inflammatory medication. No pharmacological therapy that exhibits convincing disease-modifying efficacy has been approved by regulatory agencies[3,5]. Thus, there is an urgent need to explore disease-modifying OA drugs (DMOADs) that can alleviate, halt, or even reverse the development of OA.

Although OA is a disorder of the whole joint, the progressive destruction of cartilage extracellular matrix (ECM) is considered as its hallmark. The dense ECM mainly comprises type II collagen (COL2A1) and aggrecan (ACAN)[6], and is essential for the biomechanical properties of cartilage, which provides critical elastic support to disperse pressure and shear stress as joints move. The ECM is synthesized by unique organized cells called chondrocytes, which maintain cartilage homeostasis via an equilibrium between anabolism and catabolism. Therefore, we hypothesize that modifying the cartilage ECM structure via key targets in the anabolic or catabolic processes of chondrocytes would be a promising therapeutic strategy for OA.

Dismutase 3 (SOD3) is abundant in the cartilage ECM, and is markedly decreased after OA development[7,8]. SOD3 functions as a major scavenger of superoxide anions in extracellular spaces, which are involved in the pathogenesis of OA[7]. Thus, SOD3 may have a vital role in maintaining cartilage ECM homeostasis through reactive oxygen species (ROS) control. In the present study, screening of 2320 compounds identifies a candidate of DMOAD, which upregulates SOD3. However, whether the activation of SOD3 can stimulate ECM synthesis under pathological conditions remains unknown.

Thus, in this study, we aim to determine (1) whether the found agent can promote cartilage ECM structure generation, and if so, (2) whether SOD3 is the molecular target of BNTA and upregulation of SOD3 in chondrocytes stimulates cartilage ECM synthesis.

## Results

**Screening for the candidate of DMOAD.** The 2320 structurally diverse molecules, which came from nine different functional classes, were screened (Fig. 1a). Candidates of DMOADs were screened via alcian blue staining and verified by RT-PCR. From the primary screening, 191 candidates were chosen as initial hit compounds after alcian blue staining (Fig. 1b). Then, 124 candidates remained after we excluded 67 compounds that have already been studied (Supplementary Table 1). To illustrate the chondrogenic potential of the selected compounds, COL2A1 and ACAN mRNA levels in human OA chondrocytes were examined after incubation with the above agents. Finally, ten candidates were chosen as final targets (Fig. 1b).

**Identification of a candidate of DMOAD in vitro.** N-[2-bromo-4-(phenylsulfonyl)-3-thienyl]-2-chlorobenzamide (BNTA, Fig. 1c) was identified as a strong chondrogenic agent based on the reproducibility and magnitude of its effect among the ten compounds. The characterization data of BNTA was shown in Supplementary information. Proteoglycan staining of ATDC5 cells increased dramatically after BNTA treatment for 5 d (Fig. 1d). In human OA chondrocytes, upregulation of the expression levels of ECM-related genes COL2A1, ACAN, proteoglycan 4 (PRG4), and SRY-box 9 (SOX9) was observed upon BNTA treatment (0.01–10 μM), suggesting retention of the chondrocyte phenotype (Fig. 1f). Meantime, we observed that SOX9 protein was markedly elevated after treated with BNTA

compared with vehicle (Fig. 1h). We next explored whether BNTA upregulated ECM-related genes in rat chondrocytes, in which OA was induced using interleukin 1 beta (IL1β). We observed that treatment with BNTA could increase Col2a1, Acan, Prg4, and Sox9 mRNA levels, with maximum effects around 0.1 μM (Fig. 1g). Moreover, BNTA remarkably increased the COL2A1 and SOX9 protein levels (Fig. 1i). We then evaluated BNTA therapeutic efficacy in comparison to Glucosamine sulfate (GS). Increased Acan and Sox9 mRNA levels were observed after BNTA treatment compared with GS in IL1β-induced rat OA model (Supplementary Fig. 1).

No toxicity was observed with BNTA treatment at 0.01–10 μM in human OA chondrocytes at 1 d, 3 d, 5 d, and 7 d, as determined using an alamar blue cell viability assay (Fig. 1e). Moreover, viability of cells was not affected after BNTA treatment at 0.01–10 μM in rat primary chondrocytes at 1 d, 3 d, 5 d, and 7 d (Supplementary Fig. 2).

**Modulation of ECM generation stimulated by BNTA ex vivo.** To investigate whether BNTA could modulate ECM generation by stimulating anabolic metabolism during OA degeneration, OA cartilage explants were cultured in the presence or absence of BNTA for 2 or 3 w to test its efficacy. As revealed in Fig. 2a, after 2 w of growth in culture, the proteoglycan content, as determined by safranin O-fast green staining, was dramatically increased in the BNTA group compared with that in the vehicle group, especially at 0.1 μM in terms of the integrated optical density (IOD) value for the proteoglycan content and the proteoglycan staining area percent (Fig. 2b, c). Meanwhile, after 3 w of treatment, the proteoglycan content was largely rescued in the BNTA group at 0.01–1 μM compared with that in the vehicle group (Fig. 2a–c). Immunohistological staining showed that BNTA obviously enhanced the type II collagen, and reduced the cartilage oligomeric matrix protein (COMP) and type X collagen (representative hypertrophy markers) contents after 3 w of treatment (Fig. 2d).

Furthermore, RNA was extracted and RT-PCR was performed to clarify the mRNA expression levels of anabolic and inflammatory biomarkers after BNTA treatment. As shown in Fig. 2e, the mRNA levels COL2A1, ACAN, PRG4, SOX9, and heparan sulfate proteoglycan 2 (HSPG2) were significantly elevated after exposure to BNTA compared with those in the vehicle group at 3 w. Meantime, the mRNA levels interleukin 6 (IL6) and C-C motif chemokine ligand 2 (CCL2) were decreased in the BNTA-treated group compared with those in the vehicle group. BNTA also increased the content of sulfated glycosaminoglycan (GAG) in the OA cartilage explants, and promoted cell proliferation (Fig. 2f). In brief, BNTA effectively modulated cartilage ECM generation and protected OA explants against degeneration through inducing anabolic response and inhibited inflammatory process.

**Attenuation of OA progression by BNTA treatment in rats.** Next, we sought to investigate whether intra-articular injection of BNTA could attenuate OA progression developed after anterior cruciate ligament transection (ACLT) in rats. According to experiments in vitro, we adopted BNTA at a series of dosages for the in vivo study. We observed that BNTA treatment (0.015, 0.15, and 1.5 mg kg$^{-1}$) markedly inhibited articular cartilage erosion and rescued the proteoglycan (as assessed by safranin O-fast green and alcian blue staining) and type II collagen (as detected by immunohistochemistry staining) content relative to vehicle-treated ACLT controls at 4 w (Fig. 3a). This was further confirmed by the Osteoarthritis Research Society International (OARSI) scores, which were significantly decreased in the

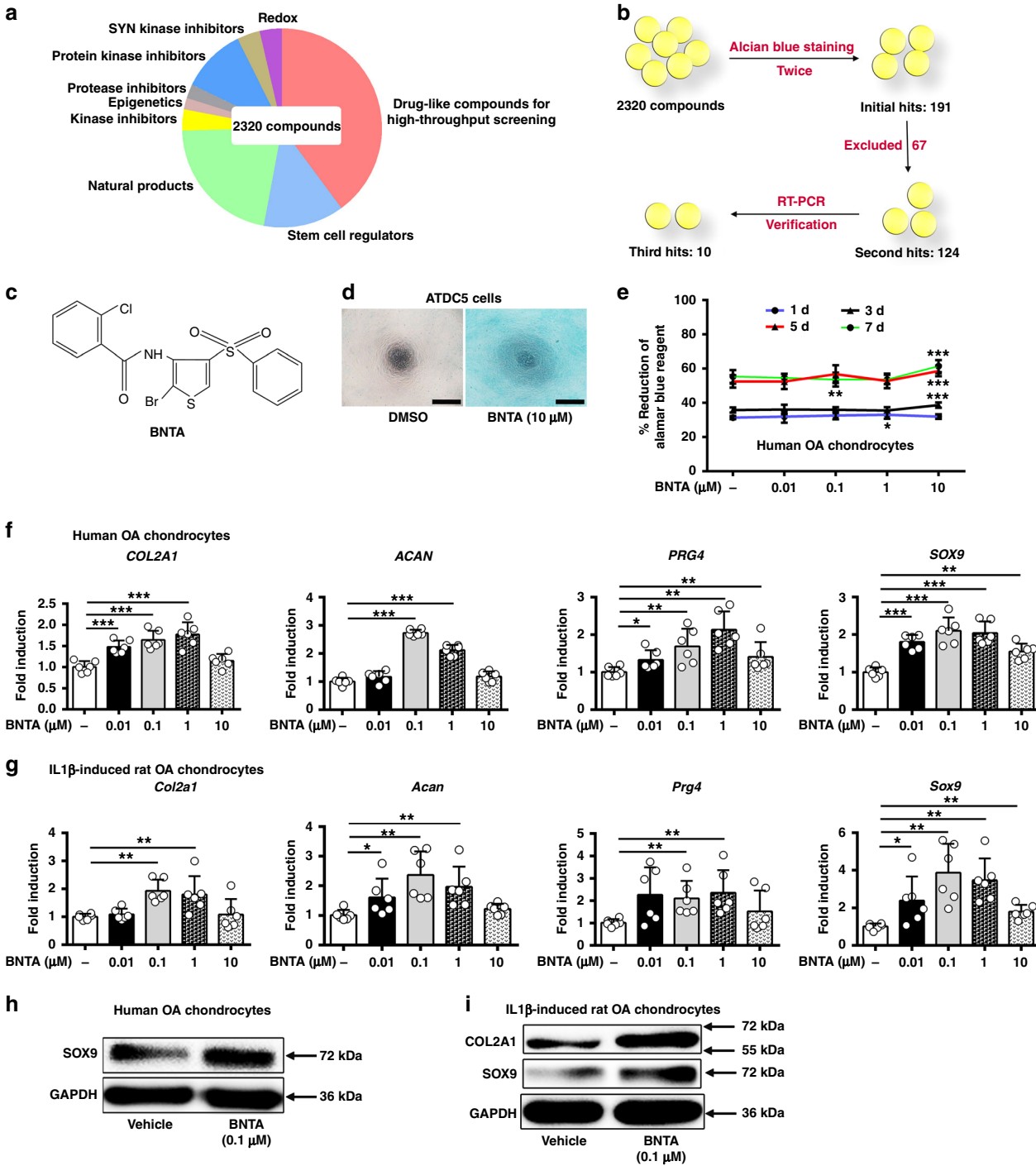

**Fig. 1** BNTA was identified as a strong chondrogenic inducer that increased anabolism of chondrocytes in vitro. **a** Pie chart of the small compounds library used for screening in different functional groups. **b** Schematic diagram of screening system using alcian blue staining and reverse transcription polymerase chain reaction (RT-PCR) verification. **c** Structure of BNTA. **d** Alcian blue staining of ATDC5 cells after incubation with BNTA or DMSO (vehicle) at 10 μM for 5 d ($n = 3$; three independent experiments; scale bars, 400 μm). **e** Cell viability of human osteoarthritis chondrocytes assessed using the alamar blue assay at 1 d, 3 d, 5 d, and 7 d after BNTA treatment ($n = 14$ for each group; 1 d, nonparametric test; 3 d, 5 d, and 7 d, one-way ANOVA). **f** Quantification of mRNA levels for collagen type II alpha 1 chain (*COL2A1*), aggrecan (*ACAN*), proteoglycan 4 (*PRG4*), and SRY-box 9 (*SOX9*) in human OA chondrocytes treated with BNTA for 6 h. Fold changes relative to vehicle control are shown ($n = 6$ for each group; *COL2A1*, *ACAN*, *SOX9*, one-way ANOVA; *PRG4*, nonparametric test; three independent experiments). **g** The mRNA levels of *Col2a1*, *Acan*, *Prg4*, and *Sox9* in interleukin 1 beta (IL1β, 10 ng ml$^{-1}$)-induced rat OA chondrocytes treated with BNTA for 6 h ($n = 6$ for each group; *Col2a1*, one-way ANOVA; *Acan*, *Prg4*, and *Sox9*, nonparametric test; three independent experiments). **h** The proteins levels of SOX9 and glyceraldehyde-3-phosphate dehydrogenase (GAPDH) in human OA chondrocytes treated with vehicle or BNTA (0.1 μM) for 2d. **i** The proteins levels of COL2A1, SOX9, and GAPDH were detected using western blotting assay in IL1β (10 ng ml$^{-1}$)-induced rat OA chondrocytes treated with or without BNTA (0.1 μM) for 2d. All data are shown as the mean ± standard deviation (s. d.). *$P < 0.05$, **$P < 0.01$, ***$P < 0.001$. Source data are provided as a Source Data file

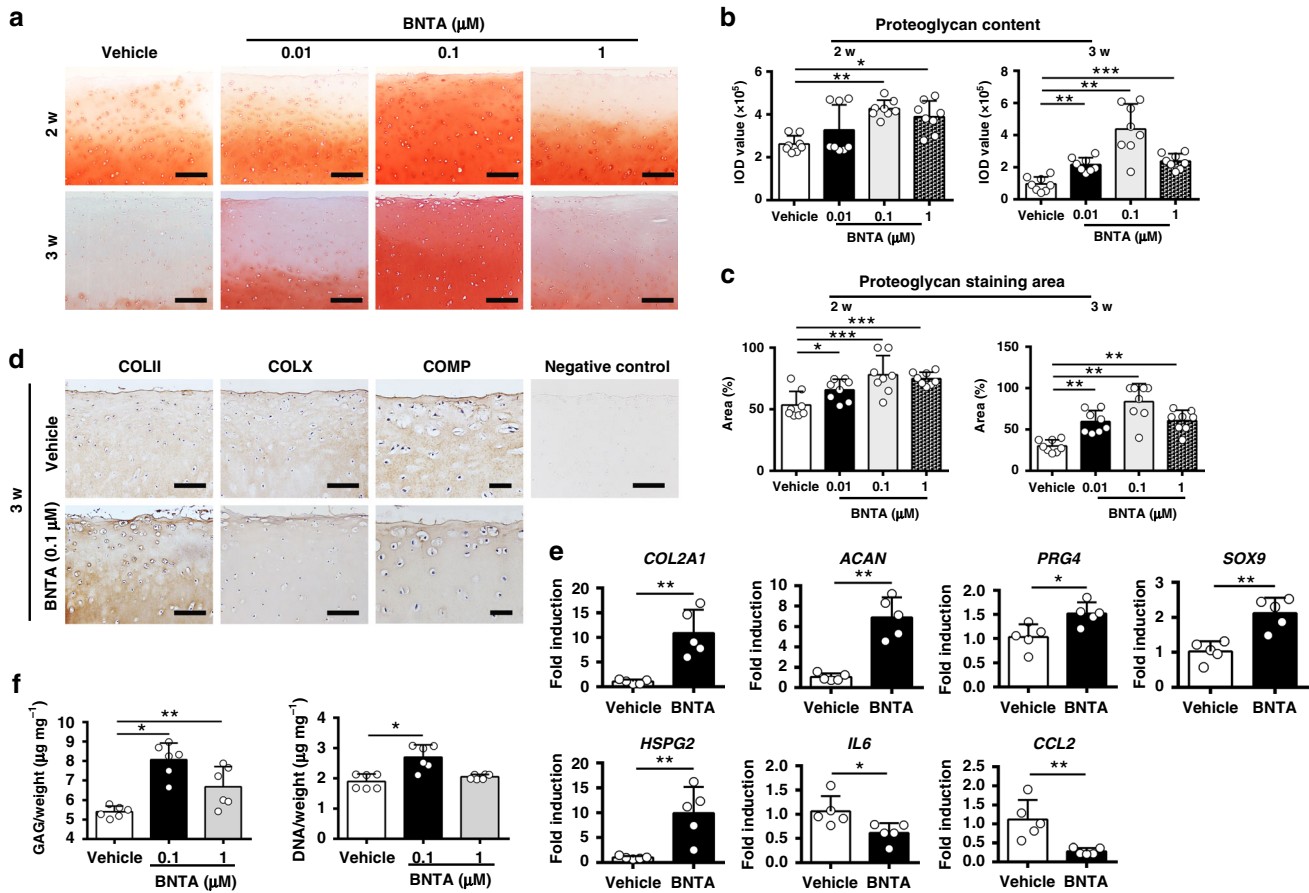

**Fig. 2** BNTA enhanced anabolism and inhibited inflammatory response in osteoarthritis cartilage explants. **a** Proteoglycan content was measured by safranin O-fast green staining in OA cartilage explants after 2 and 3 w of incubation with BNTA ($n = 8$ for each group; scale bars, 300 μm). **b** The integrated optical density value for proteoglycan content determined by safranin O-fast green staining ($n = 8$ for each group; one-way ANOVA). **c** Proteoglycan staining area (%) determined by safranin O-fast green staining ($n = 8$ for each group; 2 w, one-way ANOVA; 3 w, nonparametric test). **d** Immunohistochemistry staining of type II collagen (COLII), type X collagen (COLX), and cartilage oligomeric matrix protein (COMP) in BNTA (0.1 μM) or vehicle groups ($n = 4$; COLII, COLX, scale bars, 150 μm; COMP, scale bars, 50 μm). **e** The mRNA levels of anabolic and inflammatory biomarkers in OA cartilage explants analyzed by reverse transcription polymerase chain reaction (RT-PCR) after BNTA treatment (0.1 μM) for 3 w ($n = 5$ for each group). proteoglycan 4 (*PRG4*), SRY-box 9 (*SOX9*), and interleukin 6 (*IL6*), unpaired two-tailed Student's *t*-test; collagen type II alpha 1 chain (*COL2A1*), aggrecan (*ACAN*), heparan sulfate proteoglycan 2 (*HSPG2*), and C-C motif chemokine ligand 2 (*CCL2*), nonparametric test. **f** glycosaminoglycan (GAG) and DNA content per wet weight was assessed by 1, 9-dimethylmethylene blue (DMMB) and Hoechst 33258 assays after vehicle or BNTA treatment for 2 w, separately ($n = 6$ for each group; nonparametric test). Data are shown as the mean ± standard deviation (s. d.). *$P < 0.05$, **$P < 0.01$, ***$P < 0.001$. Source data are provided as a Source Data file

BNTA-treated ACLT rats compared with those in the vehicle-treated ACLT controls, suggesting BNTA-treated knee joints exhibited fewer osteoarthritic changes than vehicle-treated knees (Fig. 3b). We next confirmed the efficacy of BNTA injection to treat trauma-induced OA rats at 8 w. As Fig. 3c, d shown, BNTA treatment markedly reduced trauma-induced cartilage degeneration compared with that observed in the vehicle-treated group, as illustrated by increased safranin O-fast green, alcian blue, and immunohistochemistry staining for type II collagen, and decreased OARSI scores. Functional tests were applied for assessing OA alleviation effect after BNTA treatment. We observed that BNTA local administration for ACLT rats significantly reduced OA-induced pain at 4 w and 8 w post-surgery, as assessed using hot plate tests (Fig. 3e) and weight bearing tests (Fig. 3f).

Furthermore, mRNA levels of anabolic (*Col2a1*, *Acan*, *Sox9*, and *Prg4*), inflammatory (*Ccl2*, *Il11*, and *Il6*), and catabolic (ADAM metallopeptidase with thrombospondin type 1 motif (*Adamts*)1, *Adamts5*, matrix metalloproteinase (*Mmp*)3, and *Mmp*13) markers in cartilage tissue were assessed at 4 w. Cartilage

retention after BNTA treatment was confirmed by increases in ACLT-decreased mRNA levels of anabolic genes, along with decreases in catabolic and inflammatory genes expression levels compared with those in the vehicle-treated group (Fig. 4a). Similar results were observed in ACLT-treated and sham rats at 8 w (Fig. 4b).

A nanoindentation test was carried out to determine the biomechanical properties of the articular cartilage in the knee joints at 4 w. Vehicle-treated group showed a significantly lower elastic modulus and hardness compared with that in the sham group. The BNTA group exhibited a higher elastic modulus and hardness compared with those in the vehicle-treated group. Moreover, the load–depth curve revealed consistent results (Fig. 4c).

We examined the effect of intra-articular injection of BNTA on osteophyte development and subchondral bone remodeling in ACLT rats using micro-CT. BNTA remarkably decreased the osteophyte formation compared with vehicle in the medial tibial plateau of ACLT rats at 4 w and 8 w (Fig. 4d, e). Moreover, BNTA treatment suppressed the tibial subchondral bone remodeling

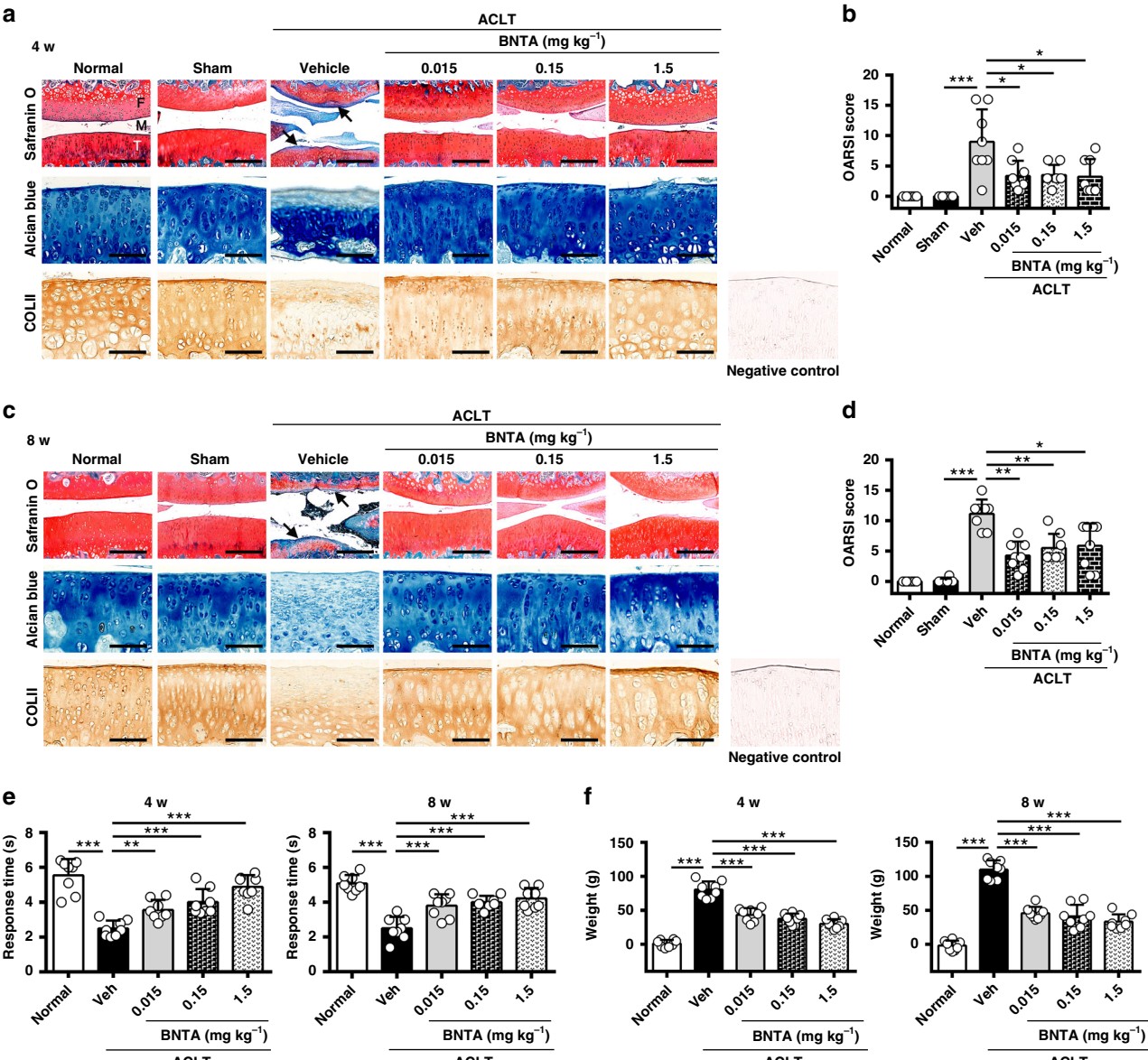

**Fig. 3** BNTA attenuated post-traumatic osteoarthritis development after intra-articular injection for 4 and 8 w. **a**, **b** Representative images of safranin O-fast green, alcian blue, immunohistochemistry staining of type II collagen (COLII), and the corresponding Osteoarthritis Research Society International (OARSI) scores from normal, sham, veh (vehicle)-treated, and BNTA-treated rats at 4 w ($n = 8$ for each group; nonparametric test; M, meniscus; F, femur; T, tibia; safranin O-fast green staining, scale bar, 200 μm; alcian blue and immunohistochemistry staining, scale bar, 100 μm). **c**, **d** Representative images of safranin O-fast green, alcian blue, immunohistochemistry staining of COLII, and OARSI scores in normal, sham, vehicle, and BNTA rats at 8 w ($n = 8$ for each group; nonparametric test; safranin O-fast green staining, scale bar, 200 μm; alcian blue and immunohistochemistry staining, scale bar, 100 μm). **e** Pain response times when rats were placed on the 55 °C hot plate meter at 4 w and 8 w post-surgery ($n = 8$ for each group, one-way ANOVA). **f** The difference between the weight placed on the contralateral sham (left) hindlimb and the weight placed on the anterior cruciate ligament transection (ACLT, right) one at 4 w and 8 w post-surgery ($n = 8$ for each group, one-way ANOVA). Each data point represents one individual rat. All data are shown as the mean ± standard deviation (s. d.). *$P < 0.05$, **$P < 0.01$, ***$P < 0.001$. Source data are provided as a Source Data file

with decreased trabecular bone volume per total volume (BV/TV) and trabecular bone pattern factor (Tb. Pf) compared with vehicle group post ACLT at 4 w and 8 w (Fig. 4d–f), as assessed by micro-CT and the corresponding quantitative analysis.

Meanwhile, the toxicity of BNTA in rats was evaluated in vivo, including anatomic features of the heart, liver, kidney, lung, and spleen, and biochemical assays. We observed that the anatomy of heart, liver, kidney, lung, and spleen of BNTA-treated ACLT rats appeared normal at 8 w, as determined by hematoxylin and eosin (H&E) staining (Supplementary Fig. 3a). In addition, serum urea, $K^+$, and $Na^+$ were normal in the BNTA group compared with the

vehicle group, as detected by the biochemical assays (Supplementary Fig. 3b, c). Therefore, we concluded BNTA at the adopted dosages showed no toxicity to rats.

**Analysis of cartilage transcriptome after BNTA treatment**. To screen for the molecular target of BNTA, we performed RNA sequencing in the samples of cartilage tissues of knee joints treated with BNTA or vehicle. We found that 885 genes were differentially expressed after BNTA treatment compared with that in the vehicle-treated samples (Supplementary Fig. 4a). Gene

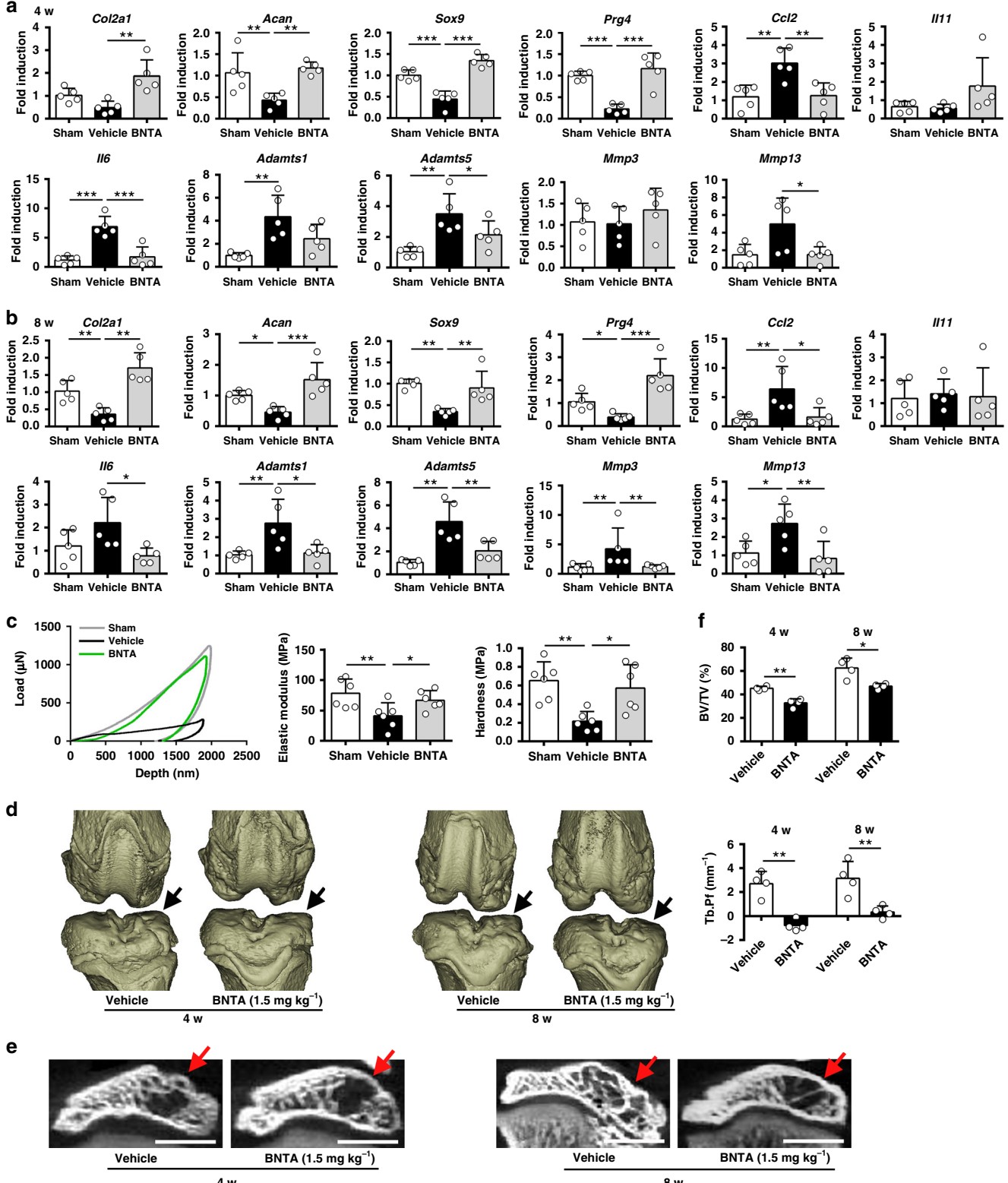

ontology (GO) analysis indicated that multicellular organismal development was upregulated, while terms including innate immune response, collagen fibril organization, inflammatory response, and aging were downregulated (Fig. 5a). Pathway analysis revealed that glycosaminoglycan biosynthesis-heparan sulfate/heparin, hedgehog signaling pathway, and ECM-receptor interaction were significantly elevated. Meanwhile, ECM-receptor interaction, PI3K-Akt signaling pathway, and rheumatoid

arthritis were downregulated (Fig. 5b). ECM-receptor interaction was significantly enriched according to pathway activity network (Fig. 5c).

Different K-cores were used to identify core regulatory genes involved in BNTA treatment. As shown in Table 1, we found that SOD3, which was most possibly related to OA development, achieved a high ranking among the top 22 genes in terms of different K-cores after excluding genes with low abundance and

**Fig. 4** BNTA attenuated post-traumatic rat osteoarthritis development. **a**, **b** Quantification of mRNA levels for collagen type II alpha 1 chain (*Col2a1*), aggrecan (*Acan*), SRY-box 9 (*Sox9*), proteoglycan 4 (*Prg4*), C-C motif chemokine ligand 2 (*Ccl2*), interleukin (*Il*)11, *Il6*, ADAM metallopeptidase with thrombospondin type 1 motif (*Adamts*)1, *Adamts5*, matrix metalloproteinase (*Mmp*)3, and *Mmp13* in articular cartilage obtained from sham, vehicle, and BNTA-treated (1.5 mg kg$^{-1}$) rats at 4 w and 8 w (a, *Col2a1*, *Acan*, *Sox9*, *Prg4*, *Adamts5*, *Ccl2*, *Il11*, *Il6*, and *Mmp3*, one-way ANOVA; *Adamts1* and *Mmp13*, nonparametric test; **b** *Acan*, *Sox9*, *Prg4*, *Il11*, and *Mmp13*, one-way ANOVA; *Col2a1*, *Ccl2*, *Il6*, *Adamts1*, *Adamts5*, and *Mmp3*, nonparametric test; $n = 5$ for each group). **c** Biomechanical properties of cartilage samples from sham, vehicle, and BNTA-treated (1.5 mg kg$^{-1}$) rats at 4 w assessed using a nanoindentation test ($n = 6$ for each group; elastic modulus, one-way ANOVA; hardness, nonparametric test). **d** Three-dimensional models for rat knee joints in the vehicle and BNTA-treated (1.5 mg kg$^{-1}$) anterior cruciate ligament transection (ACLT) rats at 4 w and 8 w. Arrows showed the medial tibial plateau. **e** Representative micro-CT images of subchondral bone in the medial tibial plateau of vehicle and BNTA-treated (1.5 mg kg$^{-1}$) ACLT rats at 4 w and 8 w. Arrows indicated medial tibial plateau. Scale bar, 2 mm. **f** Quantitative micro-CT analysis of tibial subchondral bone with trabecular bone volume per total volume (BV/TV; unpaired two-tailed Student's $t$-test; $n = 4$ for each group) and trabecular bone pattern factor (Tb. Pf; unpaired two-tailed Student's $t$-test; $n = 4$ for each group) in the vehicle and BNTA (1.5 mg kg$^{-1}$) groups at 4 w and 8 w. Each data point represents one individual rat. All data are shown as the mean ± standard deviation (s. d.). *$P < 0.05$, **$P < 0.01$, ***$P < 0.001$. Source data are provided as a Source Data file

---

matrix-related genes. Thus, we hypothesized that BNTA exhibited its OA attenuation effect by activating SOD3.

**Identification of SOD3 as the target of BNTA**. To clarify whether SOD3 was the molecular target of BNTA, mRNA and protein levels of SOD3 were firstly assessed. We observed that the SOD3 protein level in vehicle-treated ACLT rats was decreased compared with that in the normal and sham groups, but was elevated in the ACLT rats treated with BNTA for 4 and 8 w, as determined by immunohistological staining (Fig. 6a, Supplementary Fig. 4b). Similar results were observed in human OA explants (Fig. 6b). We then observed the *Sod3* mRNA levels were significantly elevated after exposure to BNTA in cartilage tissue of the rat OA model at 4 w and 8 w (compared with that in the vehicle-treated rats); *SOD3* mRNA levels were also elevated in human OA explants and rat OA chondrocytes after BNTA incubation, respectively (Fig. 6c, Supplementary Fig. 4c). While *Sod1* and *Sod2* mRNA levels remained unchanged in rat OA chondrocytes after treated with BNTA (Supplementary Fig. 4d). The protein level of SOD3 was also increased after BNTA incubation, as determined by a western blotting assay in rat OA chondrocytes (Supplementary Fig. 4e).

We then evaluated whether SOD activities and superoxide anions contents changed when treated with BNTA in rat chondrocytes. BNTA significantly increased IL1β-reduced extracellular and intracellular SOD activities in rat primary chondrocytes (Fig. 6d). Furthermore, we detected whether extracellular superoxide anions content changed in rat primary chondrocytes when incubated with IL1β or BNTA. As shown in Fig. 6e, the extracellular superoxide anions content was elevated after IL1β incubation, while significantly declined when incubated with BNTA at 2 d and 3 d. MitoSOX Red, which is specifically oxidized by superoxide anions, but not other ROS or reactive nitrogen species, was used to confirm the scavenging effect of BNTA-derived increases in SOD3 on superoxide anions. We observed the superoxide anions content in chondrocytes was significantly increased after exposure to IL1β, while was remarkably decreased after BNTA incubation as illustrated by fluorescence staining and the corresponding quantitative analysis (Fig. 6f, g). ROS in the extracellular spaces of knee joints were detected with in vivo animal imaging system CRI after luminol intravenous injection. We observed that chemiluminescence signal of the ACLT-induced knee joint was clearly evident in the vehicle group, while absent in the BNTA-treated group, which meant that local ROS production in knee joints was decreased after BNTA incubation (Supplementary Fig. 4f).

We next assessed whether SOD3 had a vital role in cartilage ECM generation via its knockdown or overexpression. We found

all three small interfering RNA targeting *Sod3* (siSOD3) effectively decreased the *Sod3* mRNA levels in rat primary chondrocytes (Supplementary Fig. 4g). We observed that when *Sod3* was elevated by the stimulation of BNTA, the mRNA levels of ECM-related genes, including *Col2a1*, *Acan*, *Prg4*, and *Sox9*, were strongly increased simultaneously, while treatment with the three siSOD3s markedly decreased the expression levels of these genes in IL1β-induced rat OA chondrocytes (Fig. 7a). The siSOD3s remarkably decreased BNTA-increased SOD3, COL2A1, and SOX9 protein levels in IL1β-induced rat OA chondrocytes (Fig. 7b). Consistent results were observed using an immunofluorescence and the corresponding quantitative assay (Fig. 7f, Supplementary Fig. 5a, b). Finally, plasmid-mediated overexpression of *Sod3* resulted in enhanced expression levels of *Sod3*, *Col2a1*, *Acan*, *Prg4*, and *Sox9* (Fig. 7c, e). COL2A1 and SOX9 protein levels were also increased with plasmid-mediated overexpression of SOD3 protein in IL1β-induced rat OA chondrocytes (Fig. 7d). Thus, BNTA effectively removed superoxide anions and modulated cartilage ECM generation by targeting SOD3 in OA chondrocytes.

## Discussion

OA has long been considered as a degenerative disease with a high prevalence and burden[1–4]. However, current therapies for OA are insufficient and no convincing DMOADs exist. In this study, we reported a candidate of DMOAD, BNTA, which significantly enhanced anabolic metabolism in OA chondrocytes, and rescued the decrease of main ECM structural molecules, such as type II collagen and ACAN, in OA cartilage explants and OA cartilage tissue developed by ACLT in rats. We confirmed that BNTA exerted its effect by upregulation of SOD3, which protected chondrocytes by catalyzing the dismutation reaction of superoxide anions (Fig. 7g). Our results showed that activating SOD3 with BNTA facilitated cartilage ECM synthesis in OA chondrocytes, OA cartilage explants, and rat OA model. Our findings not only defined the compound BNTA, which had not been applied on any diseases, as a promising DMOAD, but also manifested a distinct molecular target; i.e., SOD3-mediated OA alleviation through cartilage ECM generation.

To date, a wide variety of DMOADs targeting anabolism or catabolism of OA progression have been tested in experimental or clinical studies. An experimental study reported that kartogenin can modify OA by stimulating mesenchymal stem cells into chondrocytes by regulating CBFb-RUNX1[9]. TD-198946 prevented and repaired cartilage degeneration by inducing chondrogenic differentiation of progenitors by targeting Runx1[10]. However, it is unknown whether these agents have been subjected to further research. In clinical studies, agents such as chondroitin

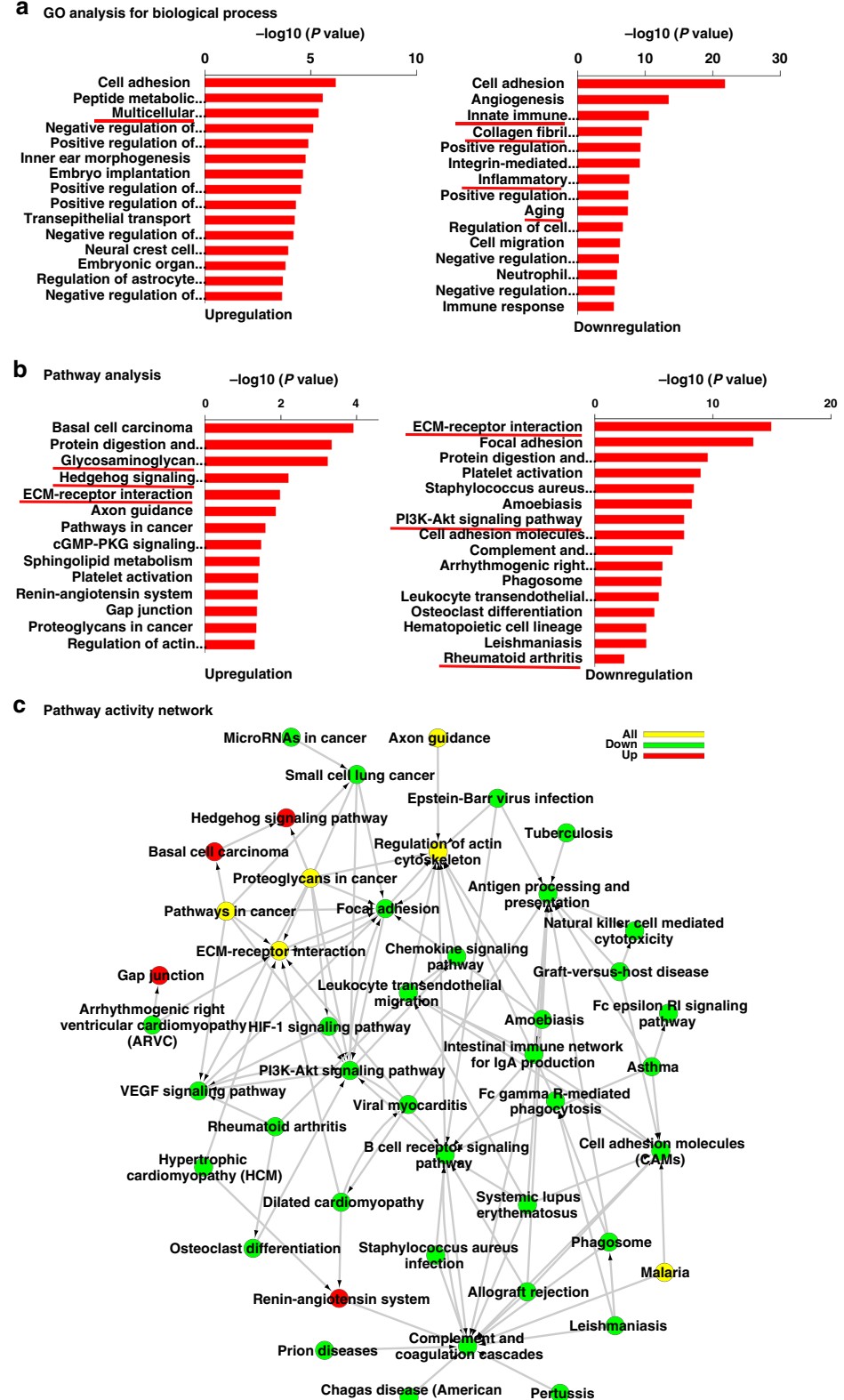

**Fig. 5** Cartilage transcriptome determined by RNA sequencing. **a** Gene ontology (GO) analysis for differentially expressed genes of cartilage transcriptome in rat anterior cruciate ligament transection (ACLT) model treated with BNTA or vehicle for 4 w. **b** Pathway analysis for differentially expressed genes from the rat cartilage transcriptome. **c** Pathway activity network construction using pathway terms (P < 0.05)

**Table 1 Differentially expressed genes ranked in front**

| AccID | K-core case | K-core control | DifKcore | Description |
|---|---|---|---|---|
| Fth1 | 44 | 16 | 28 | ferritin, heavy polypeptide 1 |
| Apoe | 44 | 18 | 26 | apolipoprotein E |
| Angptl2 | 44 | 22 | 22 | angiopoietin-like 2 |
| Pcsk6 | 44 | 22 | 22 | proprotein convertase subtilisin/kexin type 6 |
| Crim1 | 44 | 23 | 21 | cysteine rich transmembrane BMP regulator 1 (chordin like) |
| Sod3 | 36 | 16 | 20 | superoxide dismutase 3, extracellular |
| Scin | 44 | 24 | 20 | scinderin |
| Tnc | 44 | 24 | 20 | tenascin C |
| LOC100365008 | 42 | 24 | 18 | rCG54858-like |
| Thbs2 | 44 | 27 | 17 | thrombospondin 2 |
| Clu | 44 | 27 | 17 | clusterin |
| Thbs1 | 42 | 26 | 16 | thrombospondin 1 |
| Mst4 | 16 | 36 | −20 | serine/threonine protein kinase 26 |
| Ssc5d | 19 | 40 | −21 | scavenger receptor cysteine rich family, 5 domains |
| Susd5 | 19 | 40 | −21 | sushi domain containing 5 |
| Mfi2 | 19 | 40 | −21 | antigen p97 (melanoma associated |
| Hapln1 | 19 | 40 | −21 | hyaluronan and proteoglycan link protein 1 |
| Clec3a | 16 | 40 | −24 | C-type lectin domain family 3, member A |
| Mapk8ip1 | 16 | 40 | −24 | mitogen-activated protein kinase 8 interacting protein 1 |
| Aoc3 | 16 | 40 | −24 | amine oxidase, copper containing 3 |
| Gas1 | 25 | 51 | −26 | growth arrest specific 1 |
| Ctsh | 22 | 51 | −29 | cathepsin H |

sulfate[11,12], doxycycline[13], glucosamine hydrochloride[11], and SM04690[14] were utilized to treat OA. Some of these agents have slowed the structural degradation or alleviated symptoms in patients with OA. However, no existing pharmacological therapeutic intervention has been approved by regulatory agencies, suggesting that none have convincingly disease-modifying efficacy[3,5]. In this study, we found a compound, BNTA, which showed a promising potential for OA alleviation by modulating cartilage generation.

In this study, we confirmed that activating SOD3 in chondrocytes promoted ECM structural molecules synthesis, which suggested SOD3 as a potential and vital target for DMOADs development. Several studies have focused on the possible role of SOD3 in rheumatoid arthritis. Overexpression of SOD3 ameliorated rheumatoid arthritis in rats by reducing the production of pro-inflammatory cytokines[15,16], while the absence of SOD3 led to more severe arthritis compared with that in the wild type[17]. Therefore, SOD3 plays an important part in rheumatoid arthritis. There has been little research on the influence of SOD3 on OA, although certain amounts of SOD3 were observed within cartilage tissue, and its level profoundly decreased with OA progression[7,8]. Our research demonstrates that SOD3 can modulate cartilage ECM synthesis to halt or even reverse OA progression through ROS control.

SOD is pivotal antioxidant enzyme that functions as a superoxide anions scavenger by catalyzing the dismutation reaction of superoxide anions, producing first hydrogen peroxide and then water, rather than transformed into highly aggressive compounds, such as peroxynitrite and hydroxyl radicals[18]. Excessive ROS, such as superoxide anions, peroxynitrite, or hydroxyl radicals, cause direct or indirect damage to the cartilage ECM[19], DNA[19], or hyaluronic acid[20]. Thus, restoring the ROS balance is essential for cartilage homeostasis. There are three different mammalian isoenzymes: SOD1 (CuZn-SOD, located in the cytoplasm), SOD2 (Mn-SOD, located in mitochondria), and SOD3 (EC-SOD, located in the extracellular spaces)[21,22]. All three SODs show lower mRNA and protein levels in OA cartilage compared with normal cartilage[8,23,24]. SOD1 is important in neurodegenerative disorders, which causes oxidative protein damage and elevated cell death when mutated[22,25]. The main function of SOD2 is

maintaining mitochondrial function to keep cartilage homeostasis through reducing oxidative damage[26,27]. Unlike SOD1 and SOD2, SOD3 functions in extracellular spaces, such as cartilage ECM. Thus, it is likely to be more important in protecting cartilage ECM against oxidant damage than others.

Many efforts have been made to alleviate the ROS burden in OA joints, such as using native SOD, SOD mimetics, and other antioxidants[28–32]. Data from clinical trials showed that intra-articular injection of orgotein (bovine CuZn-SOD) exhibited some promise for OA treatment[29]; however, its use was limited due to immunological reactions[25]. SOD mimetic, an Mn$_{II}$-based agent, M40403, has been used to protect against rheumatoid arthritis[30,33]. No related study has been focused on OA alleviation. Other agents, such as tempol[34], a radical scavenger, and curcuminoids with strong antioxidant capacity[35], prevented joints from developing OA or alleviated the symptoms of patients with OA; however, it is not clear whether this research has been extended further.

The limitation is that this is a preliminary report and it is expected that the details of the interaction between BNTA and ECM generation require further research and will be published in a future paper.

In brief, the primary finding of this work was the identification of a candidate of DMOAD, BNTA, which facilitated cartilage structural molecule synthesis on chondrocytes by activating SOD3. Moreover, modulating SOD3 markedly enhanced cartilage ECM synthesis, making it a promising drug target for OA treatment.

## Methods

**High-throughput screening of small compound libraries.** Libraries comprising 2320 natural and synthetic small compounds were purchased from the National Compound Resource Center (Shanghai, China) and TargetMol (Boston, MA, USA). For primary screening, an image-based high-throughput screening method was established using ATDC5 cells (a mouse chondrogenic cell line) in a 96-well format. ATDC5 cells were treated with each molecule at 10 μM for 5 d by completely replacing the medium on the second day. Chondrogenesis was assessed by alcian blue staining, which stained the cartilage-specific matrix component, proteoglycan (Supplementary Fig. 6a). Chondrogenic stimulators, transforming growth factor-β3 (TGF-β3) and insulin-transferrin-selenium (ITS), were used to verify the efficacy of the primary screening[36], in which enhanced proteoglycan staining of ATDC5 cells indicated a positive result (Supplementary Fig. 6b). To

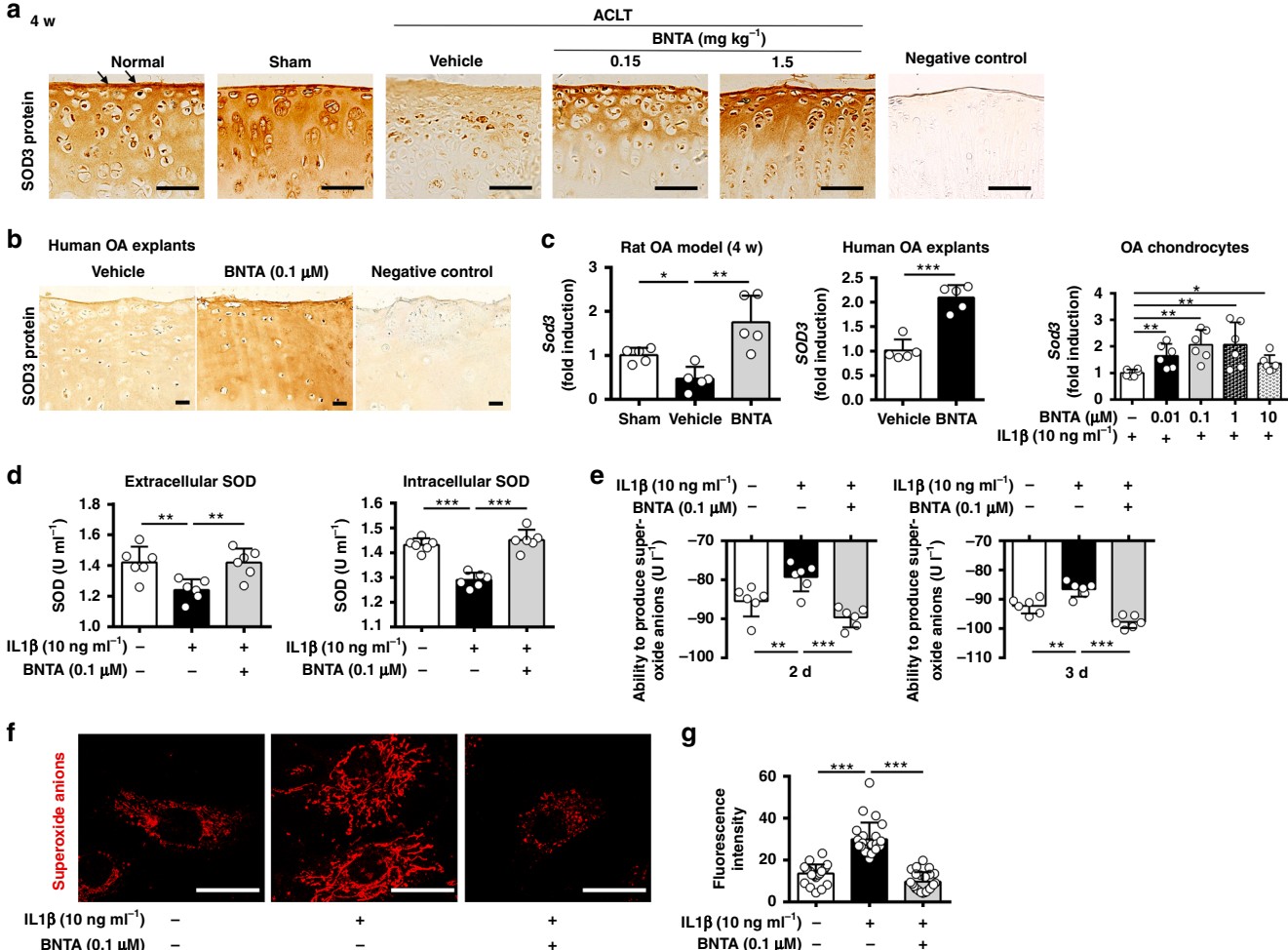

**Fig. 6** Superoxide dismutase 3 (SOD3) was the target of BNTA. **a** Representative images of immunostaining for SOD3 protein in normal, sham, vehicle, and BNTA-treated (0.15, 1.5 mg kg$^{-1}$) rats at 4 w (brown, arrows; scale bar, 50 μm). **b** Immunostaining for SOD3 protein in human osteoarthritis cartilage explants after BNTA (0.1 μM) or vehicle incubation for 3 w (scale bar, 50 μm). **c** Quantification of mRNA levels for *Sod3* in articular cartilage tissue of rat OA model after vehicle or BNTA (1.5 mg kg$^{-1}$) treatment for 4 w (nonparametric test; $n = 5$ for each group), *SOD3* in human OA cartilage explants after vehicle or BNTA (0.1 μM) treatment for 3 w (unpaired two-tailed Student's *t*-test; $n = 5$ for each group), and *Sod3* in interleukin 1 beta (IL1β, 10 ng ml$^{-1}$)-induced rat OA chondrocytes after BNTA or vehicle treatment for 6 h (nonparametric test; $n = 6$ for each group). **d** The extracellular and intracellular SOD activities in rat primary chondrocytes after treated with IL1β (10 ng ml$^{-1}$) or BNTA (0.1 μM) for 3 d (one-way ANOVA; $n = 6$ for each group). **e** The ability to produce superoxide anions of culture media in IL1β (10 ng ml$^{-1}$)-induced rat OA chondrocytes incubated with BNTA (0.1 μM) for 2 d and 3 d, which was used to assess the extracellular superoxide anions content (one-way ANOVA; $n = 6$ for each group). **f, g** Representative staining images and quantitative assay of superoxide anions in IL1β (10 ng ml$^{-1}$)-induced rat OA chondrocytes incubated with BNTA (0.1 μM) for 2 d, as assessed with MitoSOX Red staining (one-way ANOVA; $n = 24$ for each group). Scale bar, 25 μm. Data are shown as the mean ± standard deviation (s. d.). *$P < 0.05$, **$P < 0.01$, ***$P < 0.001$. Source data are provided as a Source Data file

illustrate the chondrogenesis protective potential of the selected compounds, human OA chondrocytes were treated with each compound at 10 μM for 6 h. The mRNA levels of *COL2A1* and *ACAN* were then examined using a reverse transcription polymerase chain reaction (RT-PCR) assay. If the mRNA expression levels were elevated, the compounds were regarded as chondrogenic inducers (Supplementary Fig. 6c).

**Isolation, culture of chondrocytes, and ATDC5 cells**. Human OA chondrocytes were isolated from cartilage fragments that were dissected from knee joint cartilage of OA patients discarded at the time of total joint replacement surgeries, with the approval of the Human Ethics Committee of Peking University Third Hospital. We have complied with all relevant ethical regulations for work with human participants. And the patients' informed consent was obtained. Primary rat chondrocytes were isolated from cartilage fragments dissected from femoral heads and femoral condyles of Sprague–Dawley (SD) rats weighing 80 g.

The cartilage fragments were minced and digested with 0.2% type II collagenase at 37 °C for 4 h. The cells were resuspended in Dulbecco's modified Eagle's medium (DMEM; Gibco, CA, USA) containing 10% fetal bovine serum (FBS; HyClone, Logan, UT, USA). ATDC5 cells were cultured with DMEM/F12 (Gibco) containing

5% FBS. All cells were maintained in a humidified incubator containing 5% CO$_2$ at 37 °C.

**Culture of human OA cartilage explants**. OA cartilage explants were harvested from femoral condyles of total knee arthroplasty patients. Briefly, the cartilage explants were cut into pieces of approximately 1 mm$^3$ in volume. The explants were then dispended into DMEM containing 10% FBS supplemented with BNTA (0.01–1 μM) or DMSO (vehicle), which was changed every 2 or 3 d. After 2 or 3 w of incubation, cartilage explants were collected for histological assays, mRNA expression levels determination, and 1, 9-dimethylmethylene blue (DMMB; Sigma, USA) assays.

**Induction of OA model and intra-articular injection of BNTA**. A post-traumatic OA model was induced by ACLT performed on male SD rats weighing 80 g. Briefly, under general anesthesia, the anterior cruciate ligament of the right knee was transected. A sham operation was performed on the contralateral knee with no ligament transection. We randomly divided the rats into six groups: normal, sham-operated, ACLT-operated treated with vehicle (physiological saline), and treated with BNTA (0.015, 0.15, 1.5 mg kg$^{-1}$), respectively. For intra-articular injection,

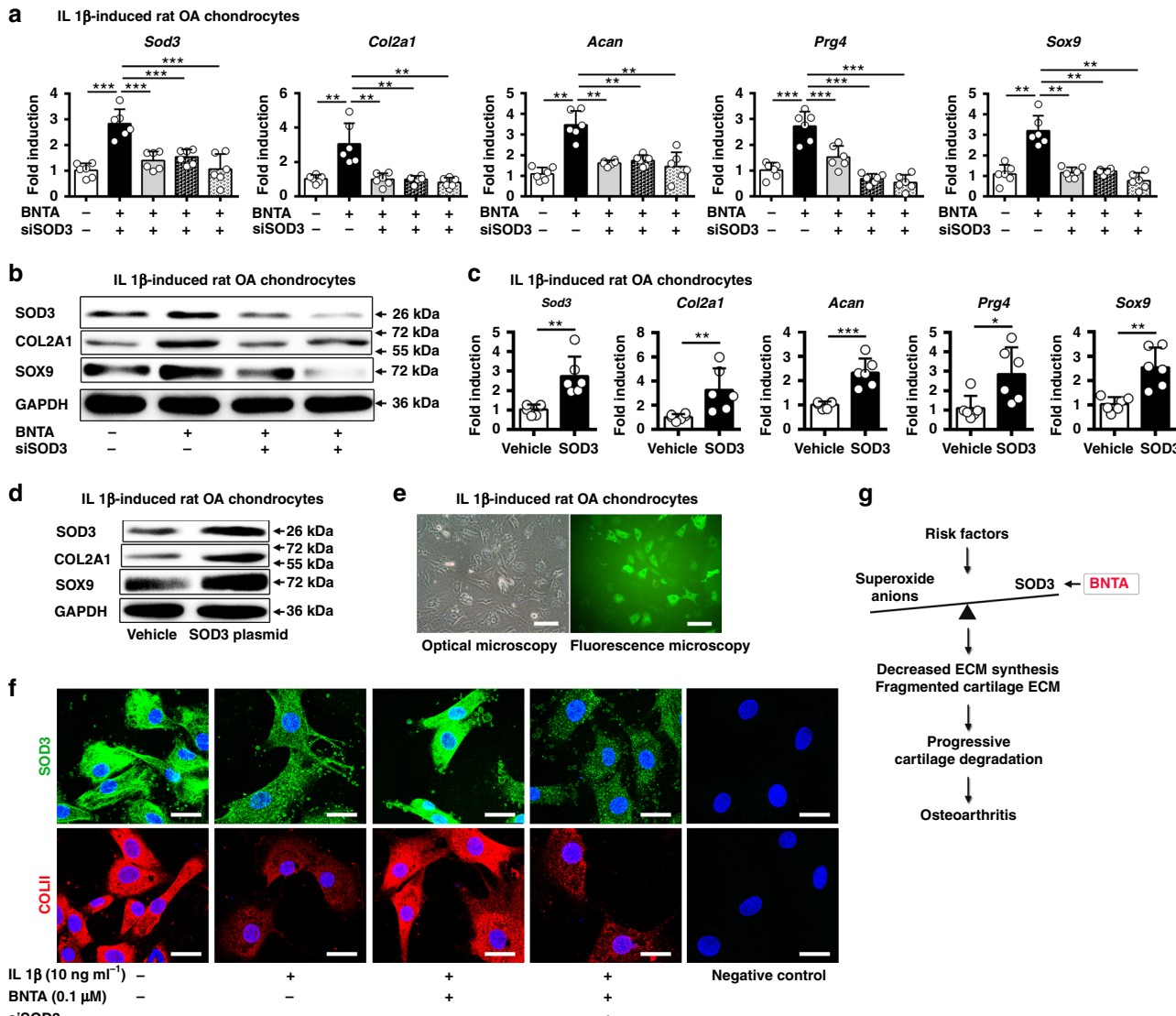

**Fig. 7** Identification of superoxide dismutase 3 (SOD3) as the target of BNTA. **a** *Sod3*, collagen type II alpha 1 chain (*Col2a1*), aggrecan (*Acan*), proteoglycan 4 (*Prg4*), and SRY-box 9 (*Sox9*) mRNA levels in interleukin 1 beta (IL1β, 10 ng ml$^{-1}$)-induced rat osteoarthritis chondrocytes cultured with BNTA (0.1 μM) or three siSOD3s for 6 h, respectively ($n = 6$ for each group; *Sod3*, *Prg4*, one-way ANOVA; *Col2a1*, *Acan*, *Sox9*, nonparametric test). **b** The proteins levels of SOD3, COL2A1, SOX9, and glyceraldehyde-3-phosphate dehydrogenase (GAPDH) in IL1β (10 ng ml$^{-1}$)-induced rat OA chondrocytes treated with BNTA (0.1 μM) or siSOD3s for 2d. **c** *Sod3*, *Col2a1*, *Acan*, *Prg4*, and *Sox9* mRNA levels in IL1β (10 ng ml$^{-1}$)-induced rat OA chondrocytes treated with vehicle or SOD3 plasmid for 2 d ($n = 6$ for each group; *Sod3*, *Acan*, and *Sox9*, unpaired two-tailed Student's t-test; *Col2a1* and *Prg4*, nonparametric test). **d** The proteins levels of SOD3, COL2A1, SOX9, and GAPDH were evaluated in IL1β (10 ng ml$^{-1}$)-induced rat OA chondrocytes treated with or without SOD3 plasmid for 2 d. **e** Images of IL1β-induced rat OA chondrocytes under optical and fluorescence microscopy after incubation with SOD3 plasmid for 2 d (scale bar, 400 μm). **f** Immunofluorescence staining for SOD3 and COLII in rat primary chondrocytes cultured with IL1β (10 ng ml$^{-1}$), BNTA (0.1 μM), or siSOD3 for 2 d. Scale bar, 25 μm. **g** The working model of how BNTA protects against OA development by activating SOD3. Data are shown as the mean ± standard deviation (s. d.). *$P < 0.05$, **$P < 0.01$, ***$P < 0.001$. Source data are provided as a Source Data file

BNTA was dissolved in physiological saline. Rats were given an intra-articular injection (100 μl) of BNTA or vehicle twice a week following surgery for 4 or 8 w, respectively. Ethical approval was received from the Animal Care and Use Committee of Peking University Health Science Center. We have complied with all relevant ethical regulations for animal testing and research. Animal experiments were conducted in accordance with appropriate international guidelines.

**RNA extraction and RT-PCR analysis**. Total RNA was isolated using the TRIzol reagent (Invitrogen, Carlsbad, CA, USA). Purified RNA (2 μg) was reverse-transcribed using a RevertAid First Strand cDNA Synthesis Kit (Thermo Fisher Scientific, Boston, MA, USA). Real-time RT-PCR was performed with the Applied Biosystems StepOnePlus Real-Time PCR System (Foster City, CA, USA). Relative gene expression levels were expressed as fold changes, calculated by the $2^{-\Delta\Delta CT}$ formula. Values were normalized to the expression of glyceraldehyde-3-phosphate

dehydrogenase (*GAPDH*) or 18S ribosomal RNA (*Rn18s*) mRNA levels. Information on the primer sequences is available upon request.

**Histological assessment**. The whole knee joints of rats were excised and fixed in 10% neutral buffered formalin for 3 d. The specimens were then decalcified for 2 d, and dehydrated in a graded ethanol series. Afterwards, the samples were embedded in paraffin, cut into 5 μm–thick sections, and stained with safranin O-fast green. For the human OA cartilage explants, histological evaluation procedures were performed without decalcification. The samples were stained with safranin O-fast green (Solarbio, Beijing, China), alcian blue (Solarbio), and subjected to immunohistological staining using antibodies recognizing type II collagen (Abcam, Camrbidge, MA, USA; 1:200), type X collagen (Gene Tex, TX, USA; 1:200), COMP (Gene Tex; 1:100), and SOD3 (Santa Cruz Biotechnology, CA, USA; 1:200). The OARSI grading system was used to score histopathologic changes in osteoarthritic cartilage[37,38].

**Cell viability assay with alamar blue**. The cell viability of chondrocytes after BNTA incubation was assessed using the alamarBlue™ Cell Viability Assay Reagent (Thermo Fisher Scientific). Cells were seeded in 96-well plates at 40,000 cells ml$^{-1}$, and maintained in culture medium supplemented with a graded BNTA series for 1 d, 3 d, 5 d, and 7 d. Then 10 μl of alamar blue reagent was added into each well and the plates were incubated in an incubator at 37 °C, 5% CO$_2$ for 4 h. Fluorescence at an excitation wavelength at 540 nm and an emission wavelength at 590 nm was measured. The background signal was determined using the negative control of medium alone without cells. The % reduction of alamar blue reagent was calculated using the fluorescence readings according to manufacturer's instructions.

**Western blotting assay**. Cells were lysed with Radioimmunoprecipitation assay (RIPA) lysis buffer, separated by SDS polyacrylamide gel electrophoresis (PAGE), and transferred to polyvinylidene fluoride (PVDF) membrane. PVDF membranes were incubated with primary antibodies overnight at 4 °C, incubated with secondary antibodies at room temperature for 1 h, and visualized using the BIO-RAD ChemiDoc XRS + system. Proteins were analyzed with antibodies recognizing COL2A1 (Abcam; 1:2000), SOX9 (Abcam; 1:3000), SOD3 (Santa Cruz Biotechnology; 1:100), and GAPDH (Zsjqbio, Beijing, China; 1:1000).

**RNA sequencing for the cartilage transcriptome**. Cartilage tissues were collected from medial and lateral femoral condyles of ACLT-operated knee joints treated with BNTA (1.5 mg kg$^{-1}$) or vehicle twice a week for 4 w, respectively. We performed RNA sequencing analysis with NovelBrain Cloud Analysis Platform. Briefly, total RNA was extracted from cartilage tissue using Trizol reagent (Invitrogen). The cDNA libraries were then constructed for each pooled RNA sample using the VAHTSTM Total RNA-seq (H/M/R). Differential gene and transcript expression analysis of RNA-seq were examined using the TopHat and Cufflinks[39]. HTseq[40] was used to count gene and lncRNA counts. Meantime, FPKM method was used to determine the gene expression. We applied DESeq algorithm to calculate the differentially expressed genes. Significant analysis was performed using the P-value and false discovery rate (FDR) analysis[41]. At the same time, differentially expressed genes were identified with: fold change >2 or fold change <0.5, FDR <0.05. Furthermore, GO analysis was performed to facilitate elucidating the biological implications of the differentially expressed genes, including biological process (BP), cellular component (CC), and molecular function (MF)[42]. The GO annotations from NCBI (http://www.ncbi.nlm.nih.gov/), UniProt (http://www.uniprot.org/), and the Gene Ontology (http://www.geneontology.org/) were downloaded. Fisher's exact test was applied to identify the significantly influenced GO categories. Pathway analysis was used to identify the significantly influenced pathways on which the differentially expressed genes have affected according to the Kyoto Encyclopedia of Genes and Genomes (KEGG) database[43]. Fisher's exact test was used to select the significantly influenced pathway. And the threshold of significance was defined by P value[44]. Pathway activity network was constructed using Cytoscape[45] for graphical representations of enriched biological pathways with significance ($P < 0.05$), including upregulated and downregulated ones. Finally, we used the method of co-expression analysis to focus on molecular target of BNTA on the gene level. The degree and K-core values of each significantly differentially expressed gene were obtained by calculating the Pearson correlation coefficient between genes. The importance of each gene for the phenotype modification was determined accordingly (the greater the degree and K-core values, the greater the co-expression ability of the indicated gene). Namely, higher ranked genes played more important roles in the whole gene network for phenotype modification than lower ranked ones. Combined with the specific functions of these genes, the molecular target of BNTA was identified.

**Hot plate test**. The hot plate test was applied to analyze the pain response in joints. The normal, vehicle, and BNTA (0.015, 0.15, and 1.5 mg kg$^{-1}$)-treated ACLT rats were placed on the hot plate meter (UGO BASILE srl, VA, ITALY) at 55 °C. The response time was recorded when the hindlimb responses, such as shaking, jumping, or licking, appeared. Rats were taken out if the hindlimb response time exceeded 30 s, avoiding scalds. Each rat was measured for three times. And the observers were blinded to the experiment.

**Weight bearing test**. The weight distribution of hind paws of rats was measured using the incapacitance tester (UGO BASILE srl). When testing, rats were standing inside in the chamber with one paw on one transducer. The duration time was set for 9 s. The results were shown as the difference between the weight placed on the contralateral sham (left) hindlimb and the weight placed on the ACLT (right) one. Measurements were taken for three times for each rat. The observers were blinded to the animal group.

**Nanoindentation test**. Biomechanical analysis of rat cartilage tissue was carried out using an in situ nanomechanical test system (TI-900 TriboIndenter, Hysitron, Minneapolis, MN, USA). Cartilage samples were collected from femoral condyles of the sham, vehicle, or BNTA-treated (1.5 mg kg$^{-1}$) groups of ACLT rats ($n = 6$) at 4 w. PBS solution was used to maintain cartilage hydration. The indentation cycle consisted of a 10 s peak load, 2 s hold, and another 10 s unload. The

maximum indentation depth was 2000 nm. Hardness and elastic modulus were determined from the load–depth curve.

**Micro-CT analysis for knee joints**. Micro-CT analysis was applied for detection of osteophyte development and subchondral bone remodeling in rat knee joints. Intact knee joints were obtained and removed the surrounding soft tissue, such as the skin and muscles, in the ACLT-induced rat OA model after exposure to vehicle and BNTA (1.5 mg kg$^{-1}$) for 4 w and 8 w. Samples ($n = 4$ for each group) were scanning the micro-CT (Siemens, Inveon MM Gantry). Three-dimensional model was reconstructed using Mimics Research software. The histomorphometric analysis was performed in the longitudinal images of the tibial subchondral bone. Moreover, tibial subchondral bone was analyzed using Inveon Research Workplace software (Siemens), including trabecular BV/TV and Tb. Pf.

**Biochemical analysis of GAG and DNA content**. The wet weights of the OA cartilage explants were determined ($n = 6$ per group). After grinding, explants were digested overnight in papainase (125 μg ml$^{-1}$) at 60 °C. The GAG content was measured using a DMMB assay. Lysates (20 μl) were mixed with 200 μl of DMMB working solution for 30 min at room temperature. The absorbance was then measured at 525 nm. Chondroitin sulfate (Sigma) was used as a standard. The DNA content was determined using the Hoechst 33258 assay (Beyotime Biotechnology, Beijing, China). Briefly, 20 μl of lysate was mixed with 200 μl of Hoechst 33258 working solution and incubated at 37 °C for 1 h. Absorbance was determined at 360 nm for excitation and at 460 nm for emission. Calf thymus DNA (Sigma) was used as the standard. The GAG or DNA content was shown as micrograms of GAG or DNA per milligram of wet weight.

**SOD activity detection**. Rat primary chondrocytes were cultured with IL1β (10 ng ml$^{-1}$) or BNTA (0.1 μM) for 3 d. The culture media were replaced at 2 d. Rat chondrocytes and culture media were obtained for SOD activity detection. Briefly, 20 μl of sample solution or ddH$_2$O were mixed with 200 μl of WST working solution. A volume of 20 μl of dilution buffer or 20 μl of enzyme working solution were then mixed thoroughly with the mixtures. The plates were maintained for 20 min at 37 °C. The absorbance at 450 nm was read using a microplate reader. Finally, SOD activity (inhibition rate %) was calculated using the equation. According to standard curve line, SOD activity (U ml$^{-1}$) was obtained.

**MitoSOX Red staining**. Rat primary chondrocytes were treated with IL1β (10 ng ml$^{-1}$) or BNTA at 0.1 μM for 2 d. Then, 1 μM MitoSOX Red (Thermo Fisher Scientific) diluted in HBSS/Ca/Mg was applied and incubated with the cells for 10 min at 37 °C. Superoxide anions, the predominant ROS produced by chondrocytes, were detected using confocal microscopy. Images were analyzed using LAS_X software (Flexera software LLC). The fluorescence intensity was quantified at least 6 images.

**Extracellular superoxide anions detection**. Rat primary chondrocytes were maintained with IL1β (10 ng ml$^{-1}$) or BNTA at 0.1 μM for 3 d, for which culture media were changed at 2 d. Extracellular superoxide anions were detected using superoxide anions detection kits (Nanjing Jiancheng Bioengineering Institute, Nanjing, China), which simulated the reaction system of astragalus and astragalus oxygenase. Culture media at 2 d and 3 d were obtained and mixed with superoxide anions detection kits for 40 min at 37 °C. The chromogenic agents were then added into the mixture standing for 10 min at room temperature. Finally, absorbance at 550 nm was measured by the microplate reader. The ability to produce superoxide anions (U l$^{-1}$) was calculated by the (OD$_{samples}$ − OD$_{control}$) / (OD$_{control}$ − OD$_{Standard}$) × standard concentration (0.15 mg ml$^{-1}$) × 1000 ml formula, which was used to measure the superoxide anions content. The ddH$_2$O was regarded as the negative control, with the ability to produce superoxide anions of 0 U l$^{-1}$ after calculation. Meantime, the Vitamin c (Vc) was considered as the standard with the ability to produce superoxide anions of −150 U l$^{-1}$.

**In vivo imaging ROS**. To test whether ROS in the extracellular spaces declined after BNTA treatment in ACLT rats, luminol (Sigma; 2.5 mg for each rat) was applied in the knee joints after vehicle or BNTA (1.5 mg kg$^{-1}$) treatment at 8 w with intravenous injection. Knee joints of each rat were imaged immediately using in vivo animal imaging system CRI (Maestro2, USA).

**Immunofluorescence analysis**. Cultured chondrocytes were rinsed in PBS, and then fixed with 10% neutral buffered formalin for 30 min at room temperature. Triton X-100 (Beyotime Biotechnology) was used to penetrate the cell membrane for 5 min, and donkey serum (Beyotime Biotechnology) was applied to block nonspecific binding sites for 1 h. Cultured cells were incubated with primary antibodies against type II collagen (Abcam; 1:200) and SOD3 (Santa Cruz Biotechnology; 1:100) at 4 °C overnight. The cells were then washed with PBS three times, and then incubated with fluorescein isothiocyanate (FITC)-conjugated anti-rabbit IgG (Abcam; 1:1000) or anti-mouse IgG (Biolegend, USA; 1:200) for 1 h. Nuclei were stained with Hoechst 33258 for 5 min. Finally, the samples were rinsed with PBS and visualized using confocal microscopy. LAS_X software (Flexera

software LLC) was applied to quantify the fluorescence intensity of rat chondrocytes from at least 6 images.

**Statistical analysis**. All data are shown as the mean ± standard deviation (s.d.). Each data point in vivo represents an individual rat. Analysis was performed using the SPSS 18.0 statistical software (IBM Corp). Statistical significance ($P < 0.05$) was calculated using unpaired two-tailed Student's $t$-tests (two groups), one-way ANOVA (homogeneity of variance, three or more groups), or nonparametric test (uneven variance).

## Data availability

All data generated or analyzed during this study are included in this published article (and its supplementary information files). The source data underlying Figs. 1e-i, 2b-c, e-f, 3b, d-f, 4a-c, f, 6c-e, g, 7a, c and Supplementary Figs. 1, 2, 3b-c, 4c-d, f, 5 are provided as a Source Data file. The RNA sequencing data has been deposited at Gene Expression Omnibus (GEO) under accession ID GSE128093. The source data files have been released at figshare [https://figshare.com/] under accession Digital Object Identifier (DOI) [10.6084/m9.figshare.7813868, 7813877, 7813901, 7813907, 7813910, 7813916, 7813799, 7813808, 7813835, 7813844, and 7813865].

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

## Acknowledgements

We thank the funding of National Natural Science Foundation of China (grant number 81330040, 81802153), Beijing Natural Science Foundation (grant number 7171014, 7182175), and Beijing New-star cross cooperation project of Science and Technology (grant number Z171100001117133).

## Author contributions

Y.A., J.W., Y.S., X.H. and J.C. conceived the project and designed the experiments. Y.S., X.H., F.Z., W.S., B.R., H.Y., Z.Li., Z. Liu., Q.L., X.D., X.F. and J.Z. performed experiments. Y.S., X.H., J.C., X.Z., P.Y. and Q.L. analyzed and interpreted the data and wrote the manuscript. Y.A. and J.W. supervised the project.

## Additional information

**Competing interests:** The authors declare no competing interests.

