## [Peer Review File · Nature Communications]

Reviewers' comments:

Reviewer #1, preclinical osteoarthritis research expert (Remarks to the Author):

Osteoarthritis (OA) is the most common form of age-related degenerative joint disease, but there is currently no effective disease-modifying therapy. This manuscript reports a candidate of novel disease-modifying OA drug (DMOAD), BNTA, which attenuates osteoarthritis. The mechanism that the authors seem to be proposing is that upregulation of SOD3 by BNTA could promote ECM-related anabolism in OA chondrocytes and thereby stimulates cartilage regeneration. Although these studies define a novel DMOAD, I do feel there are some issues that need to be addressed however.

Major comments

1. OA is a whole-joint disease characterized by not only cartilage destruction but also other whole-joint pathological changes such as osteophyte development and subchondral bone remodeling. This study focused on only cartilage degeneration/regeneration. Analysis of other whole-joint pathological changes might strengthen this manuscript.
2. The effects of BNTA are not dose dependent in many experimental conditions, including human OA chondrocytes (Fig. 1f), IL-1 β -treated rat OA chondrocytes (Fig. 1g), OA cartilage explants (Fig. 2a-c), and post-traumatic OA models (Fig. 3a-d). This raise question whether BNTA could be an effective DMOAD.
3. Technically, the n numbers for most analysis are 3. I thought that N=3 is not sufficient for statistical analysis.
4. The authors identified SOD3 as a target of BNTA. However, a report by Koike et al (Sci. Rep. 2015, 5: 11722) indicated that SOD2, but not SOD1 and SOD3, are downregulated in post-traumatic OA model. This issue should be discussed. Analysis of BNTA effects on SOD1 and SOD2 might be helpful for this issue. Additionally, the images in Fig. 5d (superoxide anions) and Fig. 5h (SOD3 and Coll-II) are not clear. It is hard to see any differences. The authors should present SOD enzyme activities in response to BNTA treatment in Fig. 5d.
5. The quality of Safranin-O staining images appears to be poor in Fig. 3a (especially ACLT-vehicle). More representative images are necessary.
6. Fig. 1f: BNTA concentration used to detect SOX9 are different from other experimental conditions.
7. Fig 3: What concentration of BNTA was used in the Fig. 3e and 3f? MMP3 (in addition to MMP13 and ADAMTS5) is also a critical catabolic effector molecule. Please show mRNA levels of MMP3.

Minor comments

1. Scale bars are not clear in all figures.
2. Text in Fig. 4 is unreadable.
2. Duplicated data were used in Fig. 5c and Supplementary Fig. 2c
3. Almost identical scheme of experimental results are presented in Fig. 5a and Supplementary Fig. 2b.

Reviewer #2, expert in development of new OA therapies (Remarks to the Author):

General Comments and Concerns:

The manuscript reports a novel compound BNTA that can attenuate osteoarthritis development and modulate cartilage ECM regeneration validated by in vitro cell screening and in vivo rat disease model.

In general, there are no significant flaws in experimental design, the rationale for experiments is valid. The discovery of a new small molecule and the SOD target is novel. However, there additional components needed to fully validate conclusions and reach the impact level of Nature Communications. For example, the RNA Seq data needs further analysis to support the authors' conclusion and contribute to the paper. The conclusion that SOD3 is the target of the drug and the mechanism if chondrocyte anabolic behavior is not fully supported by experimental data. In conclusion, the study presents a potentially novel OA therapeutic approach by enhancing the scavenging of reactive oxygen species scavenging using small molecules but the some conclusions are not supported by the data.

We suggest the author to further elucidate the following issue:

- 1) The RNA Seq data did not provide enough information about how cartilage transcriptome is changed with treatment of drug. Upregulated and downregulated pathway needs further analysis to reveal their difference between vehicle and treated groups. Further analysis needs to be considered to explain why the SOD3 is selected from the top 22 expressed genes.
- 2) Data is primarily gene expression (and histological). Without further protein analysis or more critically functional pain testing – the conclusions are weak.
- 3) Functional tests for OA treatment is to do behavior testing. Tests such as hot plate test, weight bearing test, and etc. need to be performed to fully assess therapeutic potential of BNTA in animal models. While functional results tend to correlate with the gene expression data presented, pain and disease modification are not always connected.
- 4) It's mentioned in the paper that SOD is a pivotal antioxidant that functions as a reactive oxygen species scavenger in extracellular spaces, and ROS balance is essential for cartilage homeostasis and OA treatment. Even though staining results on OA chondrocytes show decreased intracellular ROS level, more studies are needed to correlate SOD to extracellular ROS, and balance of ROS. The scavenging effect of BNTA itself and BNTA-derived increases in SOD3 should be differentiated.
- 5) More staining on in vivo tissue sections that show enhanced anabolic activity of chondrocytes, decreased ROS in extracellular spaces (if possible) after BNTA treatment is needed.
- 6) Critically for the conclusions, it is not appropriate to say that chondrocyte anabolism was the primary target and mechanism. Decreasing the inflammation can also support chondrocyte anabolism indirectly.

Detailed review:

Abstract

Page 1

Line 24: Cartilage degradation is not the only characteristics. Pain and functional loss are essential part pf OA characterize too.

Line 83: Specify the source of literature studying the excluded candidates.

Line 93: SOX9 is missing two data points.

Methods:

Page 385: Please report packages used for RNA Seq processing.

Figures:

Figure 1: Please add some data points on viability of cells after drug treatments.

Figure 2: Please specify the time of f plot.

Figure 5: There is no significant difference between images in d plot.

The claim SOD3 is the target of BNTA is weak. The evidence presented in the figure only proves SOD3's expression has negative correlation with BNTA. It's not directly proven. Need a more detailed molecular level proof.

Supplements: Please provide necessary negative controls for immunofluorescent and immunochemical staining, such as primary delete.

Point-to-point Reply to Referees # 1-2

Referees comments are in Arial 11; our responses are in Times New Roman 12;
revisions in manuscript are in red color.

Reviewers' comments

To Reviewer #1: We would like to thank Reviewer #1 for your precious time and efforts in reviewing our manuscript, and very much appreciate your positive appraisal of our work. In the revised version, we provide additional data and believe we have addressed all of your concerns.

Reviewer #1, preclinical osteoarthritis research expert (Remarks to the Author):

Osteoarthritis (OA) is the most common form of age-related degenerative joint disease, but there is currently no effective disease-modifying therapy. This manuscript reports a candidate of novel disease-modifying OA drug (DMOAD), BNTA, which attenuates osteoarthritis. The mechanism that the authors seem to be proposing is that upregulation of SOD3 by BNTA could promote ECM-related anabolism in OA chondrocytes and thereby stimulates cartilage regeneration. Although these studies define a novel DMOAD, I do feel there are some issues that need to be addressed however.

Major comments

1. OA is a whole-joint disease characterized by not only cartilage destruction but also other whole-joint pathological changes such as osteophyte development and subchondral bone remodeling. This study focused on only cartilage degeneration/regeneration. Analysis of other whole-joint pathological changes might strengthen this manuscript.

Response: Thank you for your insightful comments. As you pointed out, whole-joint pathological changes such as osteophyte development and subchondral bone

remodeling were important characteristics for OA. In the revised manuscript,

1) We have compared osteophyte development in the vehicle and BNTA (1.5 mg/kg)-treated ACLT rats at 4 w and 8 w based on micro-CT images. BNTA remarkably decreased osteophyte formation compared with vehicle in the medial tibial plateau of ACLT rats at 4 w and 8 w (figure 3j-k; page 6, line 171-174).

2) We have assessed subchondral bone remodeling in the knee joints of vehicle and BNTA (1.5 mg/kg)-treated ACLT rats at 4 w and 8 w using micro-CT, including histomorphometric and the corresponding quantitative analysis, such as trabecular bone volume per total volume (BV/TV) and trabecular bone pattern factor (Tb. Pf). BNTA treatment suppressed tibial subchondral bone remodeling with decreased trabecular BV/TV and Tb. Pf compared with vehicle group post ACLT at 4 w and 8 w (figure 3j-l; page 6, line 174-178).

2. The effects of BNTA are not dose dependent in many experimental conditions, including human OA chondrocytes (Fig. 1f), IL-1 β -treated rat OA chondrocytes (Fig. 1g), OA cartilage explants (Fig. 2a-c), and post-traumatic OA models (Fig. 3a-d). This raise question whether BNTA could be an effective DMOAD.

Response: Your points are well taken. Multiple concentrations of BNTA were adopted to evaluate its effect on OA chondrocytes, OA cartilage explants, and rat OA model, which showed the maximum beneficial effects around 0.1 μ M *in vitro* and 1.5 mg/kg *in vivo*. To avoid confusion, we have now deleted the description of “dose dependent” in the revised manuscript.

3. Technically, the n numbers for most analysis are 3. I thought that N=3 is not sufficient for statistical analysis.

Response: Thanks to your comment. In the revised manuscript, we have increased the n numbers of experiments to 5 (figure 2e, 3g-h, 5c, supplementary figure 4c, 4g), 6 (figure 1f-g, 5c, 5h, 5j), 8 (figure 3b, 3d), or 14 (figure 1e, supplementary figure 2), and added statistical data in our revised manuscript.

4. The authors identified SOD3 as a target of BNTA. However, a report by Koike et al (Sci. Rep. 2015, 5:11722) indicated that SOD2, but not SOD1 and SOD3, are downregulated in post-traumatic OA model. This issue should be discussed. Analysis of BNTA effects on SOD1 and SOD2 might be helpful for this issue. Additionally, the images in Fig. 5d (superoxide anions) and Fig. 5h (SOD3 and Coll-II) are not clear. It is hard to see any differences. The authors should present SOD enzyme activities in response to BNTA treatment in Fig. 5d.

Response : Thanks to your insightful suggestions, we have performed additional experiments and added the new data in the revised manuscript:

1) We have analyzed *Sod1* and *Sod2* expressions at mRNA level in IL1 β -induced rat OA chondrocytes after BNTA treatment. We observed that *Sod1* and *Sod2* mRNA levels remained unchanged after BNTA treatment (supplementary figure 4d; page 7, line 218-219).

2) We have added better representative images (superoxide anions) in figure 5f. Moreover, statistical data were added to quantify the contents of superoxide anions (figure 5g; page 7-8, line 229-235).

3) We have added more representative images (SOD3 and COLII) in figure 5m, and added the statistical data in supplementary figure 5 (page 8, line 250-252).

4) We have included SOD enzyme activities in response to BNTA treatment in figure 5d. BNTA significantly increased IL1 β -reduced extracellular and intracellular SOD activities in rat primary chondrocytes (page 7, line 222-225).

5. The quality of Safranin-O staining images appears to be poor in Fig. 3a (especially ACLT-vehicle). More representative images are necessary.

Response : Per your suggestion, we have included new safranin-O staining images in figure 3a and 3c.

6. Fig. 1f: BNTA concentration used to detect SOX9 are different from other experimental conditions.

Response: Thanks to your comments, we have performed additional experiments to detect *SOX9* expression in the missing two data points (figure 1f).

7. Fig 3: What concentration of BNTA was used in the Fig. 3e and 3f? MMP3 (in addition to MMP13 and ADAMTS5) is also a critical catabolic effector molecule. Please show mRNA levels of MMP3.

Response: Your points are well taken. In the revised version,

1) We have added the concentration of BNTA (1.5 mg/kg) in the figure legend 3g and 3h.

2) We have included the *Mmp3* mRNA level in the figure 3g and 3h. *Mmp3* mRNA levels showed no difference in the sham, vehicle, and BNTA-treated (1.5 mg/kg) ACLT rats at 4 w (figure 3g). While BNTA treatment decreased the ACLT-increased mRNA levels of *Mmp3* at 8 w (figure 3h).

Minor comments

1. Scale bars are not clear in all figures.

Response: According to your suggestions, we have redrawn scale bars in all figures.

2. Text in Fig. 4 is unreadable.

Response: According to your suggestions, we have updated figure 4 with readable data. Furthermore, figure 4d was replaced with table 1 in order to be readable.

3. Duplicated data were used in Fig. 5c and Supplementary Fig. 2c

Response: Thanks to your comments, figure 5c represented the *Sod3* mRNA levels in ACLT-induced rat OA model at 4 w. Supplementary figure 2c indicated *Sod3* mRNA levels in rat OA model at 8 w. To avoid confusion, we have now added the description in figure 5c and supplementary figure 4c in the revised manuscript.

4. Almost identical scheme of experimental results are presented in Fig. 5a and Supplementary Fig. 2b.

Response: Per your suggestion, we have provided the better representative figures in figure 5a and supplementary figure 4b.

To Reviewer #2: We would like to thank you for your precious time and efforts in reviewing our manuscript and highly appreciate your insightful comments and suggestions for our work. In the revised version, we have provided additional mechanistic insights and believe we have addressed all of your concerns.

Reviewer #2, expert in development of new OA therapies (Remarks to the Author):

General Comments and Concerns:

The manuscript reports a novel compound BNTA that can attenuate osteoarthritis development and modulate cartilage ECM regeneration validated by in vitro cell screening and in vivo rat disease model. In general, there are no significant flaws in experimental design, the rationale for experiments is valid. The discovery of a new small molecule and the SOD target is novel. However, there additional components needed to fully validate conclusions and reach the impact level of Nature Communications. For example, the RNA Seq data needs further analysis to support the authors' conclusion and contribute to the paper. The conclusion that SOD3 is the target of the drug and the mechanism if chondrocyte anabolic behavior is not fully supported by experimental data. In conclusion, the study presents a potentially novel OA therapeutic approach by enhancing the scavenging of reactive oxygen species scavenging using small molecules but the some conclusions are not supported by the data.

We suggest the author to further elucidate the following issue:

1) The RNA Seq data did not provide enough information about how cartilage transcriptome is changed with treatment of drug. Upregulated and downregulated pathway needs further analysis to reveal their difference between vehicle and treated

groups. Further analysis needs to be considered to explain why the SOD3 is selected from the top 22 expressed genes.

Response : Thanks to your insightful suggestions. Pathway analysis was used to identify the significantly influenced pathways on which the differentially expressed genes with up-regulation and down-regulation have influenced between the vehicle and treated groups according to the Kyoto Encyclopedia of Genes and Genomes (KEGG) database. Fisher's exact test was used to select the significantly influenced pathway. And the threshold of significance was defined by P value. We observed that glycosaminoglycan biosynthesis-heparan sulfate/heparin, hedgehog signaling pathway, and ECM-receptor interaction were significantly elevated. Meanwhile, ECM-receptor interaction, PI3K-Akt signaling pathway, and rheumatoid arthritis were downregulated (figure 4b).

We used the method of co-expression analysis to focus on molecular target of BNTA on the mRNA level. The degree and K-core value of each significantly differentially expressed gene was obtained by calculating the Pearson correlation coefficient between genes. The importance of each gene for the phenotype modification was then determined accordingly (the greater the degree and K-core values, the greater the co-expression ability of the indicated gene). Namely, higher ranked genes may play more important roles in the whole gene network for phenotype modification than lower ranked ones. We then detected the mRNA levels of *Crim1*, *Sod3*, *Thbs2*, *Mapk8ip1*, *Gas1*, and *Ctsh* in rat cartilage tissue of the vehicle and BNTA group, which were possibly related to OA development among the top 22 genes in terms of different K-cores. Among them, the change of SOD3 expression was mostly obvious (data not shown). Combined with the specific functions of these genes (Reply table 1), we thereby focused on SOD3 as a molecular target for BNTA, which was most possibly related to OA development.

Reply table 1. The functions of the top 22 expressed genes

AccID	Function (provided by NCBI)
Fth1	a major function of ferritin is the storage of iron
ApoE	a major apoprotein of the chylomicron

Angptl2	participate in the formation of blood vessels
Pcsk6	a member of the subtilisin-like proprotein convertase family
Crim1	may play a role in tissue development through interactions with members of the transforming growth factor beta family
Sod3	antioxidant enzymes, which may protect the brain, lungs, and other tissues from oxidative stress
Scin	a Ca(2+)-dependent actin-severing and -capping protein
Tnc	encodes an extracellular matrix protein with a spatially and temporally restricted tissue distribution
LOC100365008	function unclear
Thbs2	disulfide-linked homotrimeric glycoprotein that mediates cell-to-cell and cell-to-matrix interactions
Clu	a secreted chaperone that can under some stress conditions
Thbs1	an adhesive glycoprotein that mediates cell-to-cell and cell-to-matrix interactions
Mst4	a member of the GCK group III family of kinases, which are a subset of the Ste20-like kinases
Ssc5d	function unclear
Susd5	function unclear
Mfi2	a cell-surface glycoprotein found on melanoma cells, the iron binding function has not yet been identified
Hapln1	stabilizes cartilage proteoglycan/hyaluronic acid aggregates by binding to each component
Clec3a	function unclear
Mapk8ip1	may play a role in the SAPK mediated stress-induced cellular responses
Aoc3	localized to the adipocyte plasma membrane and to glucose transporter Glut4 containing vesicles
Gas1	plays a role in growth suppression

2) Data is primarily gene expression (and histological). Without further protein analysis or more critically functional pain testing – the conclusions are weak.

Response: Your points are well taken. In the revised manuscript,

(1) We have added the protein analysis on the model of human OA chondrocytes, IL1 β -induced rat OA chondrocytes after treated with BNTA (0.1 μ M), and SOD3 knockdown or overexpression in rat OA chondrocytes. SOX9 protein was elevated after treated with BNTA compared with vehicle in human OA chondrocytes (figure 1h, antibody against human COL2A1 was not available for western blotting; page 4, line 95-96). BNTA also remarkably increased the COL2A1 and SOX9 protein levels in IL1 β -induced rat OA chondrocytes (figure 1i; page 4, line 100-101). Furthermore, the siSOD3s remarkably decreased BNTA-increased SOD3, COL2A1, and SOX9 protein levels in IL1 β -induced rat OA chondrocytes (figure 5i; page 8, line 248-250). COL2A1 and SOX9 protein levels were increased with plasmid-mediated overexpression of SOD3 protein in IL1 β -induced rat OA chondrocytes (figure 5k; page 8, line 254-255).

(2) We have included the functional pain testing, including hot plate test and weight bearing test for assessing OA alleviation effect after BNTA treatment. We observed that BNTA local administration for ACLT rats significantly reduced OA-induced pain at 4 w and 8 w post-surgery, as assessed using hot plate test (figure 3e) and weight bearing test (figure 3f; page 5-6, line 153-156).

3) Functional tests for OA treatment is to do behavior testing. Tests such as hot plate test, weight bearing test, and etc. need to be performed to fully assess therapeutic potential of BTNA in animal models. While functional results tend to correlate with the gene expression data presented, pain and disease modification are not always connected.

Response: Thank you for your thoughtful comments. In the revised version, we have included the functional tests, including hot plate test and weight bearing test for

evaluation of OA alleviation effect. We observed that BNTA local administration for ACLT rats significantly reduced OA-induced pain at 4 w and 8 w post-surgery, as assessed using hot plate tests (figure 3e) and weight bearing tests (figure 3f; page 5-6, line 153-156).

4) It's mentioned in the paper that SOD is a pivotal antioxidant that functions as a reactive oxygen species scavenger in extracellular spaces, and ROS balance is essential for cartilage homeostasis and OA treatment. Even though staining results on OA chondrocytes show decreased intracellular ROS level, more studies are needed to correlate SOD to extracellular ROS, and balance of ROS. The scavenging effect of BNTA itself and BNTA-derived increases in SOD3 should be differentiated.

Response: Per your suggestions, in the revised version,

(1) We have analyzed the extracellular ROS content in the culture media cultured for rat primary chondrocytes treated with IL1 β (10 ng/ml) or BNTA at 0.1 μ M for 2 d or 3 d using superoxide anions detection kits assay. We found that the extracellular superoxide anions content was elevated after IL1 β incubation, while significantly declined when incubated with BNTA at 2 d and 3 d (figure 5e; page 7, line 225-229).

Furthermore, extracellular ROS in rat joints were evaluated after BNTA (1.5 mg/kg) treatment in ACLT rats at 8 w using intravenous injection of luminol. We observed that chemiluminescence signal of the ACLT-induced knee joint was clearly evident in the vehicle group, while absent in the BNTA-treated group, which meant that local ROS production in knee joints was decreased after BNTA local administration (supplementary figure 4f; page 8, line 235-240).

(2) We have compared superoxide anions contents in rat primary chondrocytes treated with IL1 β , BNTA, or siSOD3s, respectively. Decreased superoxide anions contents were observed after BNTA treatment, while significantly elevated when treated with siSOD3s in IL1 β -induced rat OA chondrocytes. Therefore, we concluded that BNTA scavenged superoxide anions through its-derived increases in SOD3 (Reply figure 1).

Rat primary chondrocytes

IL1 β (10 ng/ml)	-	+	+	+	+
BNTA (0.1 μ M)	-	-	+	+	+
siSOD3	-	-	-	+	+

Reply figure 1. Representative staining images of superoxide anions in IL1 β (10 ng/ml)-induced rat OA chondrocytes incubated with BNTA (0.1 μ M) or siSOD3s for 2 d, which were detected with MitoSOX Red staining. Scale bar, 25 μ m.

5) More staining on in vivo tissue sections that show enhanced anabolic activity of chondrocytes, decreased ROS in extracellular spaces (if possible) after BNTA treatment is needed.

Response : Thanks to your insightful comments, we have performed additional experiments on tissue staining and in vivo rat knee joints.

(1) We first tested whether anabolic activity of chondrocytes was enhanced after BNTA treatment through safranin O-fast green and alcian blue staining (representing proteoglycans) and immunohistochemical staining for type II collagen. We found that BNTA treatment (0.015, 0.15, and 1.5 mg/kg) markedly inhibited articular cartilage erosion and rescued the proteoglycan and type II collagen content relative to vehicle-treated ACLT controls at 4 w (figure 3a) and 8 w (figure 3c; page 5, line 140-144, 149-153) .

(2) To test whether ROS in the extracellular spaces declined after BNTA treatment in ACLT rats. Luminol (2.5 mg for each rat) was applied using intravenous injection. Decreased ROS content was observed in the knee joints of BNTA (1.5 mg/kg)-treated ACLT rats than vehicle-treated ones (supplementary figure 4f; page 8, line 235-240).

6) Critically for the conclusions, it is not appropriate to say that chondrocyte anabolism was the primary target and mechanism. Decreasing the inflammation can also support

chondrocyte anabolism indirectly.

Response: Thank you for your insightful suggestions. According to figure 2e and 3g-h shown, inflammatory markers (IL6 and CCL2) decreased after BNTA treatment in human OA explants and ACLT-induced rat OA models. Therefore, we have rewritten the conclusions as “We demonstrated that the DMOAD, BNTA, attenuated OA progression by facilitating cartilage ECM regeneration with increased anabolism and decreased inflammatory response in chondrocytes.” (page 2; line 36-39) and “In brief, the primary finding of this work was the identification of a novel candidate of DMOAD, BNTA, which facilitated cartilage structural molecule synthesis on chondrocytes by activating SOD3.” (page 10; line 328-330).

Detailed review:

Abstract

Page 1

Line 24: Cartilage degradation is not the only characteristics. Pain and functional loss are essential part of OA characterize too.

Response: In the revised version, we have included OA characteristic of pain and functional loss in the abstract section. Therefore, we have rewritten this sentence as “OA is characterized by pain and functional loss, and progressive degradation of the cartilage extracellular matrix (ECM), which is essential to maintain cartilage function.” (page 2, line 23).

Line 83: Specify the source of literature studying the excluded candidates.

Response: Your points are well taken. In the revised manuscript, we have specified the source of literature studying the excluded candidates in supplementary table 1

Line 93: SOX9 is missing two data points.

Response: In the revised version, we have included *SOX9* mRNA levels in the missing two data points (figure 1f).

Methods:

Page 385: Please report packages used for RNA Seq processing.

Response: Thank you for your suggestions. We performed RNA sequencing analysis with NovelBrain Cloud Analysis Platform. In the revised version, we have included the methodology of RNA Seq processing (page 13-14, line 430-461).

Figures:

Figure 1: Please add some data points on viability of cells after drug treatments.

Response: Per your suggestion, we have added viability of cells after maintained in culture medium supplemented with a graded BNTA series for 1 d, 3 d, 5 d, and 7 d. Furthermore, we have increased the n number of experiments to 14 for each group (figure 1e and supplementary figure 2; page 4, line 105-109).

Figure 2: Please specify the time of f plot.

Response: In the revised version, we have included the time of f plot in the figure legend 2.

Figure 5: There is no significant difference between images in d plot.

The claim SOD3 is the target of BNTA is weak. The evidence presented in the figure only proves SOD3's expression has negative correlation with BNTA. It's not directly proven. Need a more detailed molecular level proof.

Response: Thank you for your insightful comments. In the revised version,

(1) We have updated with the better representative images (figure 5f). Moreover, statistical data showed that BNTA significantly reduced IL1 β -induced superoxide anions contents (figure 5g).

(2) We have added the molecular level proof between SOD3's expression and BNTA using the western blot assay. As figure 5i shown, BNTA remarkably increased the SOD3, COL2A1, and SOX9 proteins levels compared with vehicle in IL1 β -induced rat OA chondrocytes, while siSOD3s remarkably decreased them (page 8, line 248-250).

Supplements: Please provide necessary negative controls for immunofluorescent and immunochemical staining, such as primary delete.

Response: Your point is well taken. Accordingly, in the revised manuscript,

(1) We have added the negative controls of immunofluorescent in figure 5m.

(2) We have included the negative controls of immunochemical staining in figure 2d, 3a, 3c, 5a, 5b, and supplementary figure 4b.

Editorial comments:

Additional concerns expressed during editorial assessment of the manuscript are the absence of data on BNTA therapeutic efficacy in a chronic disease setting and in comparison to previously reported treatments; and the lack of causation-establishing evidence in support of the proposed mechanism of action.

Response: Thank you for your insightful suggestion. In the revised version,

1) We have added the data on BNTA therapeutic efficacy in a chronic disease setting and causation-establishing evidence in support of the proposed mechanism of action as reviews' comments.

2) We evaluated BNTA therapeutic efficacy in comparison to Glucosamine sulfate (GS), which was used to modify symptoms in OA. Increased *Acan* and *Sox9* mRNA levels were observed after BNTA treatment compared with GS in IL1 β -induced rat OA model (supplementary figure 1; page 4, line 101-104).

REVIEWERS' COMMENTS:

Reviewer #1 (Remarks to the Author):

All issues are satisfactorily addressed.

Minor comments.

1. Line 198 (p.7). I can't find 'rheumatoid arthristis' in Figure 4b.
2. Line 268 (p. 9). Figure 5i might be Figure 5n.
3. Revise IL-1 β concentration "10 ng/ml" to "10 ng ml⁻¹" and BNTA concentration "mg/kg" to "mg kg⁻¹". Please check all units throughout whole manuscript.
4. Figure 5c (OA chondrocytes): please add IL-1 β (10 ng ml⁻¹) in the X axis.
5. Figure 5i, k, and l: please add "IL-1 β -induced rat OA chondrocytes" as in figure 5h and j to avoid confusion.

Reviewer #2 (Remarks to the Author):

Authors have answered in detail to the questions raised previously.

1. Authors have answered the major concern, which was to differentiate extracellular and intracellular level of SOD with BNTA treatment.
2. Authors completed behavior testing, as well as the CT images, to confirm functional restoration of the joint.
3. Authors better highlighted the importance of BNTA and SOD using siSOD study.

Authors responded to the comments and revised. Can be published.

Reviewers' comments

To Reviewer #1: We would like to thank Reviewer #1 for your precious time and efforts in reviewing our manuscript, and very much appreciate your positive appraisal of our work. In the revised version, we provide additional data and believe we have addressed all of your concerns.

Reviewer #1 (Remarks to the Author):

All issues are satisfactorily addressed.

Minor comments.

1. Line 198 (p.7). I can't find 'rheumatoid arthristis' in Figure 4b.

Response: Thank you for your comments. In the revised version, we have included the “rheumatoid arthritis”.

2. Line 268 (p. 9). Figure 5i might be Figure 5n.

Response: Thank you for pointing out our mistakes. In the revised version, we have corrected it.

3. Revise IL-1 β concentration "10 ng/ml" to "10 ng ml⁻¹" and BNTA concentration "mg/kg" to "mg kg⁻¹". Please check all units throughout whole manuscript.

Response: Thank you for your comments. We have revised all of them.

4. Figure 5c (OA chondrocytes): please add IL-1 β (10 ng ml⁻¹) in the X axis.

Response: Per your suggestion, we have added this in the X axis.

5. Figure 5i, k, and l: please add "IL-1 β -induced rat OA chondrocytes" as in figure 5h and j to avoid confusion.

Response: Per your suggestion, we have added them in figure 7.

To Reviewer #2: We would like to thank you for your precious time and efforts in reviewing our manuscript and highly appreciate your insightful comments for our work.

Reviewer #2 (Remarks to the Author):

Authors have answered in detail to the questions raised previously.

1. Authors have answered the major concern, which was to differentiate extracellular and intracellular level of SOD with BNTA treatment.

Response: Thank you for your comments.

2. Authors completed behavior testing, as well as the CT images, to confirm functional restoration of the joint.

Response: Thank you for your comments.

3. Authors better highlighted the importance of BNTA and SOD using siSOD study.

Response: Thank you for your comments.

Authors responded to the comments and revised. Can be published.

Response: Thank you for your comments.